# Broadening the Knowledge of Mexican Boletes: Addition of a New Genus, Seven New Species, and Three New Combinations

**DOI:** 10.3390/jof9121126

**Published:** 2023-11-21

**Authors:** Olivia Ayala-Vásquez, Jesús Pérez-Moreno, Juan Pablo Pinzón, Roberto Garibay-Orijel, Jesús García-Jiménez, Javier Isaac de la Fuente, Crystian Sadiel Venegas-Barrera, Magdalena Martínez-Reyes, Leticia Montoya, Víctor Bandala, Celia Elvira Aguirre-Acosta, César Ramiro Martínez-González, Juan Francisco Hernández-Del Valle

**Affiliations:** 1Edafología, Campus Montecillo, Colegio de Postgraduados, Carretera México-Texcoco Km. 36.5, Montecillo, Texcoco 56230, Estado de México, Mexico; yootspooj@gmail.com (O.A.-V.); jdelafuenteitcv@gmail.com (J.I.d.l.F.); martinezreyes.2012@gmail.com (M.M.-R.); 2Tecnológico Nacional de México, Instituto Tecnológico de Ciudad Victoria, Blvd. Emilio Portes Gil #1301Pte., Ciudad Victoria 87010, Tamaulipas, Mexico; jgarjim@yahoo.com.mx (J.G.-J.); crystianv@gmail.com (C.S.V.-B.); ramiro_mg.unam@ciencias.unam.mx (C.R.M.-G.); sanfrancisco_hv89@hotmail.com (J.F.H.-D.V.); 3Departamento de Botánica, Facultad de Medicina Veterinaria y Zootecnia, Universidad Autónoma de Yucatán, Carretera Mérida-Xmatkuil, Km 15.5, Mérida 97100, Yucatán, Mexico; juan.pinzone@correo.uady.mx; 4Instituto de Biología, Universidad Nacional Autónoma de México, Circuito Exterior s/n Ciudad Universitaria, Mexico City 04510, Mexico; rgaribay@ib.unam.mx (R.G.-O.); caguirre@ib.unam.mx (C.E.A.-A.); 5Red Biodiversidad y Sistemática, Instituto de Ecología A.C., Xalapa 91073, Veracruz, Mexico; leticia.montoya@inecol.mx (L.M.); victor.bandala@inecol.mx (V.B.)

**Keywords:** boletes, edible wild mushrooms, ethnomycology, Neotropics, subtropical ecosystems

## Abstract

Boletes are one of the most common groups of fungi in temperate, subtropical, and tropical ecosystems. In Mexico, the northern region has mainly been explored in terms of bolete diversity. This study describes a new genus and seven new species based on macromorphological, micromorphological, molecular, phylogenetic, and ecological data. *Garcileccinum* gen. nov. is typified with *G. salmonicolor* based on multigene phylogenetic analysis of nrLSU, RPB2, and TEF1, and it is closely related to *Leccinum* and *Leccinellum*. *Garcileccinum viscosum* and *G. violaceotinctum* are new combinations. *Boletellus minimatenebris* (ITS, nrLSU, and RPB2)*, Cacaoporus mexicanus* (RPB2 and ATP6), *Leccinum oaxacanum, Leccinum juarenzense* (nrLSU, RPB2, and TEF1), *Tylopilus pseudoleucomycelinus* (nrLSU and RPB2), and *Xerocomus hygrophanus* (ITS, nrLSU, and RPB2) are described as new species. *Boletus neoregius* is reclassified as *Pulchroboletus neoregius* comb. nov. based on morphological and multigene phylogenetic analysis (ITS and nrLSU), and its geographic distribution is extended to Central Mexico, since the species was only known from Costa Rica. Furthermore, *T. leucomycelinus* is a new record from Mexico. This study contributes to increasing our knowledge of boletes and expands the diversity found in Mexican forests.

## 1. Introduction

Many ecosystems thrive in Mexico ranging between tropical, subtropical, temperate, subalpine, and alpine, which develop in different soil types, weather conditions, and contrasting altitudes due to the complex orography and geological history of the country [1,2]. These conditions have produced communities where fungi can easily diversify [3,4]. Consequently, more than 2000 fungal species have been reported in the country [3]. Despite the relatively high diversity recorded in Mexico, studies are far from complete. Currently, most of the records have come from temperate forests. However, tropical and subtropical forests harbor abundant species of the families Casuarinaceae, Dipterocarpaceae, Fagaceae, Fabaceae, Mimosaceae, Nictaginaceae, Pinaceae, and Polygonaceae, which establish ectomycorrhizal symbiosis and, therefore, may potentially contain high fungal diversity. Despite this fact, in general, these forests have been scarcely studied [4,5,6]. In addition to the ecological importance of ectomycorrhizal fungi in nutrient cycling in the forests [6,7], many species play an important sociocultural and economic role, especially as a food resource for indigenous communities. These species mainly include those belonging to Amanitaceae, Agaricaceae, and Boletaceae [7,8]. In the case of Boletaceae, there are species whose annual international market value is equivalent to millions of USD, e.g., the species complex commonly known as “porcini” have an annual in-season retail market higher than USD 250 million [9]. Additionally, the use of hundreds of edible species belonging to Boletaceae, Calostomataceae, Gomphidiaceae, Hygrophoropsidaceae, Suillaceae, and Sclerodermataceae [2,9] constitutes—either as food for consumption or as an important economic income—an important survival strategy based on the multiple uses of non-timber forest products for the livelihood of people in impoverished social groups [7,10].

The Boletaceae family includes species with stipitate-pileate or hypogeous forms; they mainly have a tubular hymenophore, but sometimes, lamellate hymenophores or intermediate forms that show transitions between these two hymenophore types may be found [11,12]. Currently, this family groups seven subfamilies: Austroboletoideae, Boletoideae, Chalciporoideae, Leccinoideae, the Pulveroboletus group, Xercomoideae, and Zangioideae [12,13]. The Leccinoideae subfamily includes the genera *Chamonixia* Rolland, *Leccinum* Gray, *Leccinellum* Bresinsky and Manfr. Binder, *Octaviania* Vittad., *Rossbeevera* T. Lebel and Orihara, and *Turmalinea* Orihara and N. Maek [14]. However, recently, it has been considered that *Leccinum* and *Leccinellum* are polyphyletic [14,15,16] and that the group includes several unresolved clades, especially within the species described from Central America, e.g., the clade of *Leccinellum viscosum* (Halling and B. Ortiz) Mikšík [17] and *Leccinum violaceotinctus* B. Ortiz and T.J. Baroni, both described from Belize [18]. Approximately 90% of the fungal species of the Boletaceae family are obligate symbionts and represent between 18 and 25% of all ectomycorrhizal fungal diversity [19,20]. These species form ectomycorrhizal associations with different tree species and shrubs belonging to the families Betulaceae, Casuarinaceae, Dipterocarpaceae, Ericaceae, Fabaceae, Fagaceae, Mimosaceae, Myrtaceae, Pinaceae, Polygonaceae, and Salicaceae [20,21]. According to several authors, many of the tree species have migrated from the Holarctic region to the Neotropical region, and they carry along with them their associated symbiotic fungi [22,23,24]. Many species of Boletaceae have been described from Central and South America [25,26,27,28,29,30,31,32,33,34,35,36,37].

Mexican boletes have been studied since the 1960s, with initial records of *Heimioporus alveolatus* (R. Heim and Perr.-Bertr.) E. Horak, *Strobilomyces strobilaceus* (Scop.) Berk., *Strobilomyces strobilaceus* var. *mexicanus* R. Heim and Perr.-Bertr., and *Strobilomyces strobilaceus* var. *zapotecorum* R. Heim and Perr.-Bertr. distributed in central and southeast Mexico [38]. Since then, they have been the object of many taxonomic studies, mainly in north, central, and southeast Mexico [39,40,41,42,43,44,45,46,47,48,49,50,51,52,53,54]. In contrast, boletes from tropical ecosystems have been studied less [55,56,57]. Currently, 44 genera of the Boletaceae family have been described in Mexico [2,40] out of 104 genera described worldwide [58]. Although Boletaceae species are common in fungal catalogs from Mexico, many species lack a formal description or have only been recently included in checklists; therefore, more integrative taxonomic work is an urgent need in order to gain deeper insights into species complexity. This paper describes *Garcileccinum* as a new genus and *Boletellus minimatenebris, Cacaoporus mexicanus, Garcileccinum salmonicolor, Leccinum oaxacanum, Leccinum juarenzense, Tylopilus pseudoleucomycelinus*, and *Xerocomus hygrophanus* as new species. *Garcileccinum viscosum, G. violaceotinctum*, and *Pulchroboletus neoregius* are proposed as new combinations. Additionally, *Tylopilus leucomycelinus* is described for the first time from Mexico. Descriptions, photographs, drawings of the microscopic characteristics, and a phylogeny based on multi-locus regions (ITS, nrLSU, RPB1, RPB2, TEF1, and ATP6) are presented. Therefore, this work aims to contribute to the knowledge of bolete diversity in Mexico and in the Neotropics.

## 2. Materials and Methods

### 2.1. Samples and Morphology Studies

The studied specimens were collected during the rainy season, from June to November in the years 2016 to 2023, mainly in central and southern Mexico. Vegetation in the sampling areas corresponds to mixed conifer–broadleaf forests that extend throughout the Sierra Madre Occidental, the Sierra Madre Oriental, and the Neovolcanic Axis. Abundant tree species include mainly *Abies religiosa* (Kunth) Schltdl. and Cham., *A. hickelii* Flous and Gaussen, *A. guatemalensis* Rehder, *Alnus acuminata* Kunth, *Pinus pseudostrobus* Lindl., *P. teocote* Schltdl. and Cham., *P. ayacahuite* Ehrenb. ex Schltdl, *P. montezumae* Lamb., *P. patula* Schiede ex Schltdl. and Cham., *Quercus laurina* Humb. and Bonpl., *Q. scytophylla* Liebm. *Q. rugosa* Née, and relics of *Pseudotsuga menziesii* (Mirb.) Franco var. *glauca* (Beissn.) Franco. Additionally, cloud forest covers large areas containing *Carpinus* L., *Liquidambar* L., *Magnolia* L., *Oreopanax* Decne. and Planch., *P. chapensis* (Martínez) Andresen, *P. devoniana* Lindl., *P. leiophylla* Schltdl. and Cham, *Q. candicans* Née, *Q. laurina*, *Q. sapotiifolia* Liebm, *Q. skinneri* Benth., *Q. ocoteifolia* Liebm., *Q. uxoris* McVaugh, *Quercus* spp. [59], and *Ternstroemia* Mutis ex L.f. Morphological characteristics of basidiomes were described according to Li et al. [13]. Chemical reactions were characterized using KOH and NH_4_HO. Photographs of basidiomata were taken in situ, and vegetation types of samples were recorded. The colors for the taxonomic description were based on the Methuen Handbook [60]. Microscopic characteristics were measured with an optical microscope (Carl Zeiss GmbH 37081, Oberkochen, Germany); 30–35 basidiospores, basidia, pleurocystidia, cheilocystidia, pileipellis, and elements of both stipitipellis were measured. For basidiospores, the Q index (length/width) was calculated. The ornamentation of basidiospores was observed using a scanning electron microscope (MEB) (Hitachi Su 1510, Hitachi, Japan). The specimens were deposited in the National Herbarium of Mexico of the Institute of Biology of the National Autonomous University of Mexico (MEXU-HO), the Herbarium “José Castillo Tovar” of the Instituto Tecnológico de Ciudad Victoria (ITCV), and the Herbarium INECOL-Xalapa. 

### 2.2. DNA Extraction, PCR Amplification, and Sequencing

For DNA extraction, PCR, and sequencing, samples of dehydrated basidiomata at 48 °C were used. The DNA was extracted with the DNeasy Power-Soil (QIAGEN, Hilden, Germany) kit. Cell lysis was performed by grinding in mortar with liquid nitrogen. Four nuclear markers (ITS, nrLSU, RPB1, and RPB2) were amplified with the Platinum Taq DNA Polymerase (Invitrogen-Thermo Fisher Scientific, Waltham, MA, USA) and Taq and Load PCR Mastermix (MP Biomedicals, Santa Ana, CA, USA) enzymes in thermocycler (BIO-RAD, Hercules, CA, USA). The PCR parameters were 95 °C of initial denaturation each 4 min for 35 cycles: denaturation at 94 °C for 60 s, alignment at 54 °C for 60 s, extension at 72 °C for 60 s, and a final extension of 72 °C for 10 min. For the ITS region, we used the ITS1/STI4, nrLSU, LROR/LR5 [61], the largest genes of RNA polymerase II RPB1(fRPB1-Cr/bRPB2-6F), translation elongation translation factor 1-α (tef1) (primers TEFF and TEFR) [62], the second-largest RPB2 (RPB2 RPB2-B-F2/RPB2-B-R) subunits [63], and the subunit 6 ATPasa (primers atp6-6 and atp6-5) [64], respectively. Amplification was tested with electrophoresis in 1% agarose gel, dyed with GelRed (Biotium, Fremont, CA, USA), and developed in UVP Multidoc-It transilluminator (Analytikjena, Jena, Germany). LB charge buffer was only required for PCR products generated with Taq-Platinum. PCR products with successful amplification were cleaned with ExoSAP-IT (Thermo Fisher Scientific, Waltham, MA, USA) diluted 1:1 with ddH_2_O (double-distilled water). The reaction had a ratio of 2 μL of the diluted reagent to 3.5 μL of the PCR product; we conducted incubation in thermocycler at 37 °C for 45 min and 80 °C for 15 min. The sequencing reaction and capillary sequencing were performed as a service by the Laboratory of Genomic Sequencing of Biodiversity and Health of the Institute of Biology of the National Autonomous University of Mexico, where sequences in both directions for each sample were made using the BigDye Terminator v3.1 (Thermo Fisher Scientific, Waltham, MA, USA) chemistry.

### 2.3. Alignments, Model Selection, and Phylogenetic Analyses

The consensus sequences were compared with those in the GenBank database of the National Center for Biotechnology Information (NCBI) using BLASTN v. 2.2.19 [65]. In order to estimate the phylogenetic position of the specimens studied in this work, Bayesian inference and maximum likelihood analyses were performed on the following molecular matrices: (1) ITS, nrLSU, and RPB2 of *Boletellus* Murrill species with *Hemileccinum rugosum* G. Wu and Zhu L. Yang as outgroup (Table 1); (2) RPB2 and ATP6 of *Cacaoporus* Raspé and Vadthanarat, species, with specimens of *Cyanoboletus* Gelardi, Vizzini, and Simonini, *Chalciporus* Bataille, *Lanmaoa* G. Wu and Zhu L. Yang, and *Rubinoboletus rubinus* (W.G. Sm.) Pilát and Dermek as outgroup, according to Vadthanarat et al. [66] (Table 2); (3) nrLSU, RPB2, and TEF1 of several members of the subfamily Leccinoideae, including *Leccinum*, *Leccinellum*, the new genus *Garcileccinum*, *Rugiboletus, Ionosporus, Spongiforma, Pseudoaustroboletus,* and *Retiboletus*, with *Tylocinum* as outgroup [14,15] (Table 3) (additionally, ITS, nrLSU, and RPB2 sequences were obtained and deposited in the Genbank); (4) ITS and nrLSU of *Pulchroboletus* Gelardi, Vizzini, and Simonini species with *Alessioporus* Gelardi, Vizzini, and Simonini, as outgroup (Table 4); (5) ITS and nrLSU of *Tylopilus* P. Karst. species, with *Xanthoconium sinense* as outgroup [26] (Table 5); and (6) ITS, nrLSU, and RPB2 of *Xerocomus* Quél. species, with *Phylloporus imbricatus* as outgroup [67] (Table 6). All sequences, except for those produced for this study, were retrieved from the GenBank in an attempt to represent most taxa from those groups. All matrices were aligned using the MUSCLE algorithm [68] in MEGA v. 11 with default parameters [69] and visually inspected in BioEdit v. 7.2.5 [70] for corrections and trimming. For Bayesian inference (BI) analysis, we selected DNA substitution models according to the Akaike information criterion (AIC) using jModelTest v. 2.1.7 [71]. A majority-rule BI consensus tree was produced in MrBayes v. 3.2.6 [72,73] with the following parameters: substitution model retrieved from jModelTest v. 2.1.7, with two independent runs each running for 10 million generations sampling every 1000 generations with one cold chain and three hot chains with a temperature of 0.2 and a final burn-in fraction of 0.25. For the concatenated matrices, each region was treated as an independent partition, and the following parameters were unlinked: transition/transversion rate, state frequency, and shape. The rate model was set to “variable”, and the remaining parameters were used as default. Additionally, a maximum likelihood (ML) analysis was carried out in RAxML v. 8.2.12 [74] using the GTR+GAMMA substitution model treating each partition as independent. In addition, ML bootstrap (BS) was performed based on 1000 replicates. Both BI and ML analyses were implemented through CIPRES Science Gateway v. 3.3 [75].

## 3. Results

### 3.1. Phylogenetic Analyses

#### 3.1.1. *Boletellus* Phylogeny

The concatenated matrix of ITS, nrLSU, and RPB2 consisted of 31 accessions and 2478 positions, of which 809 (32.6%) were variable and 612 (24.7%) were parsimony-informative. The phylogram obtained from the BI analysis shows that two specimens of the new species, *B. minimatenebris*, are included in clade A (1BI/98ML), together with *B. ananiceps, B. aurocontextus*, *B. ananas*, and two unidentified specimens of *Boletellus*. One of them (HKAS122526) forms a well-supported clade with the two specimens of *B. minimatenebris* (1BI/98ML). Clade A is a sister to *B. brunoflavus* (0.97BI) (Figure 1).

#### 3.1.2. *Cacaoporus* Phylogeny

The concatenated matrix of RPB2-ATP6 genes consisted of 44 accessions and 1483 positions, of which 457 (30.8%) were variable and 371 (25.0%) were parsimony-informative. In the phylograms of both BI and ML analyses, *Cacaoporus* was recovered as monophyletic (1BI/99ML) and sister to a clade, which includes a paraphyletic *Cyanoboletus* with *Cupreoboletus* nested within it. This sister relationship is moderately well supported (1BI/79ML). A monophyletic *Lanmaoa* (0.99BI/98ML) is sister to both clades. The four sequences of *Cacaoporus mexicanus* (Figure 2) form a monophyletic species (1BI/99ML) that is sister to an unidentified specimen of *Cacaoporus* (SV0402) (1BI/95ML). The most closely related species to this clade is a monophyletic *Cacaoporus pallidicarneus*, whose sister relationship is moderately supported (71ML).

#### 3.1.3. Leccinoideae Subfamily Phylogeny

The concatenated matrix of nrLSU-RPB2-TEF1 genes consisted of 90 accessions and 2239 positions, of which 903 (40.3%) were variable and 788 (22.8%) were parsimony-informative. At the deepest nodes of the phylogram (Figure 3), a trichotomy is observed, composed of *Tylocinum griseolum* and two well-supported clades. The first one (1BI/81ML) includes the earliest divergent genera in the subfamily, *Retiboletus*, *Pseudoaustroboletus*, *Spongiforma*, *Ionosporus*, and *Rugiboletus*, and the other (1BI/100ML) includes *Spongispora temasekensis* and *Kaziboletus rufescens* at its deepest nodes, with the remaining genera in a core clade, including the monophyletic *Octaviania*, *Chamonixia*, the new genus proposed here (*Garcileccinum*), *Rossbeevera pachydermis*, the paraphyletic *Leccinellum*, and the polyphyletic *Leccinum*. Considering the type species of the genera, a *Leccinellum* s.s. clade can be recognized (type: L. *crocipodium*) (0.94BI/81ML) as well as a Leccinum s.s. clade (type: *L. aurantiacum*) (1BI/100ML). The new genus Garcileccinum (1BI/100ML) is sister to a clade formed by the *Rossbeevera pachydermis* and *Leccinellum s.s.* clade. Within *Garcileccinum*, three monophyletic species were recovered: *G. viscosum*, *G. violaceotinctum*, and the newly described species and type of the genus, *G. salmonicolor*; all of these were well supported (1BI/100ML).

#### 3.1.4. *Pulchroboletus*

The phylogenetic analysis of the genus *Pulchroboletus* (Figure 4) was performed with a concatenated matrix of the nrLSU and ITS markers of 17 accessions. The matrix had 1450 positions, of which 278 (19.2%) were variable and 154 (10.6%) were parsimony-informative. *Pulchroboletus* was found to be monophyletic with high support (1BI/100ML). Inside the genus, a dichotomy is observed: one clade is composed of one sequence of *P. rubricitrinus* (0.98BI/91ML) and a monophyletic group of five sequences of *P. roseoalbidus* (1BI/94ML). The other clade (0.95BI/82ML) is composed of three specimens of unidentified *Pulchroboletus*, a clade of *P. sclerotiorum*, and the new combination here proposed: *P. neoregius*. Both species are monophyletic, highly supported (1BI/100ML), and are sister taxa.

#### 3.1.5. *Tylopilus* Phylogeny

Two independent phylogenetic analyses were performed for *Tylopilus*, and another was performed with ITS and nrLSU, due to the missing sequences, to produce the four markers’ concatenated matrix. The two-marker matrix consisted of 105 accessions and 2168 positions, of which 1193 (55.0%) were variable and 828 (35.2%) were parsimony-informative. The phylogram (Figure 5) demonstrated low resolution and/or support at the deepest nodes. *Tylopilus balloui* is highly polyphyletic, appearing in several lineages. The specimen described here as a new species, *T. pseudoleucomycelinus*, is sister to another specimen identified as *T. balloui* (FMNH1073250) with high support (0.99BI/99ML). This clade is sister to a monophyletic *T. leucomycelinus* with good support (0.98BI/85ML). The lineage related to the type locality of *T. balloui* (USA) has good support outside this clade.

#### 3.1.6. *Xerocomus* Phylogeny

The genes (nrLSU, ITS, and RPB2) consisted of 43 accessions and 2566 positions, of which 662 (25.8%) were variable and 457 (17.8%) were parsimony-informative. The deepest nodes of the phylograms of both BI and ML show low resolution, but some well-supported clades are recovered in *Xerocomus*. The new species, *X. hygrophanus*, is sister to an unidentified specimen of *Xerocomus* sp. (MAN2011-b-MAN061) with good support (0.96BI/70ML) and separated from a clade of specimens of *X. illudens s.s.* Additionally, this clade is sister to another sequence, identified as *X. illudens* (DD9854), which is not well supported (69ML) and requires further study (Figure 6).

### 3.2. Taxonomy

***Boletellus minimatenebris*** Ayala-Vásquez and Garibay-Orijel, **sp. nov.** (Figure 7, Figure 8A–C and Figure 9A,B).

MycoBank no: MB 834539.

Etymology: from the Latin “minima” (tiny), in reference to the small basidiomata, and “tenebris” (dark), related to the dark brown to darkish basidiomata.

Holotype: Mexico, Oaxaca state, Mixistlán de la Reforma Municipality, Santa María Mixistlán Town, Mootsa’am Place, on soil, 3 October 2017, N 17°7′58.62″ W 96°5′22.26″, Ayala-Vásquez O. (ITCV1076, MEXU-HO-30119, ITS GenBank No: OR713119, nrLSU GenBank No: OR713121, RPB2 GenBank No: OQ938895).

Diagnosis: basidiomata: small; pileus surface: furfuraceous to velvety; dark brown, with a sterile margin; hymenophore: subdecurrent or adhered yellow; tubes and pores: concolorous, light yellow changing to dark blue and gray–blue when cut. Basidiospores: 13–16 (–19) × 6–7 (–8) μm, Q = 2.2 (*n* = 34), ellipsoid, golden yellow to yellowish in KOH with longitudinal striations.

Description: pileus: 10–34 mm in diameter, convex, dark brown (5F4–F8), velvety to furfuraceous; pileus surface: straight margin, sterile, decurvate. Hymenophore: subdecurrent; pores: 0.2–0.5 mm, circular to irregular, yellow (4A8), turning dark blue to blue–gray (20D6) when touched; tubes: 1–2 mm long, concolorous to the pores, turning blue–gray when cut (20D6). Pileus context: 2–4 mm thick, whitish to pale yellow, turning blue–gray when cut (21D6); context: whitish at the apex of the stipe, ruby (12C8) in the middle part, blackish brown (7E5) at the base. Stipe: 25–47 × 5–8 mm, subclavate, brown (5F4–5F6) to dark brown at the base, furfuraceous. Basal mycelium: brown (5E4). Odor: fungoid. Taste: fungoid. Chemical reaction: the pileus and stipe turn blackish brown with NH_4_OH; the hymenophore and context turn pale brown. Basidiospores: 13–16 (–19) × 6–7 (–8) μm (Q = 2.2–2.35, Qm = 2.28, *n* = 140), ellipsoid, golden yellow to yellowish in KOH with longitudinal striations, some almost verrucose, shallow, or with transverse veins or stretch marks. Basidia: 30–33 (−38) × 12–13 μm, clavate, four-spored, golden to yellowish brown in KOH, golden in Melzer’s reagent. Pleurocystidia: 50–55 (−67) × 10–11 (−12) μm, fusoid, ventricose-rostrate, golden to yellowish brown in KOH, golden in Melzer’s reagent, thick-walled. Cheilocystidia: 56–63 (−75) × 10–11(−12) μm, ampullaceous, ventricose-rostrate with a septum, brown in KOH and dark brown in Melzer’s reagent, thin-walled. Pileipellis composed of suberect hyphae, with terminal cells of 22–60 (−72) × 8–12 (−15) μm, cylindrical, fusoid, ventricose-rostrate, brown in KOH, thin-walled. Caulocystidia: 28–45 (−59) × 9–14 (−17) μm, fusoid, ventricose-rostrate, mucronate, brown in KOH, with intracellular content in Melzer’s reagent; caulobasidia: 36–45 × 8 (−15) μm, clavate, four-spored, hyaline in KOH, yellow–brown in Melzer’s reagent.

**Habitat and distribution**: solitary, growing associated with *Quercus elliptica* in *Quercus–Pinus* forests, during October; only found in Mixistlán, Oaxaca, Mexico.

**Additional specimens examined**: Mexico, Oaxaca state, Mixistlán de la Reforma Municipality, Santa María Mixistlán Town, Mootsa’am place, 3 October 2017, N 17°07′59″ W 96°05′24.91″, Ayala-Vásquez (ITCV-1087, MEXU-HO-30132, paratype); Kauxkukukm place, 3 October 2017, N 17°07′01.79″ W 96°05′24″, Ayala-Vásquez, O. (ITS GenBank no: OR713120, nrLSU GenBank no: OR713122; RPB2 GenBank no: OQ938894).

**Notes**: *Boletellus minimatenebris* is characterized by its small basidiomata compared to other species of the genus, section *Boletellus*. *Boletellus minimanebris* is characterized by dark brown pileus, yellow hymenophore that turns dark blue when cut, whitish to pale yellow context that turns blue when cut, and basidiospores 13–16 (–19) × 6–7 (–8) μm, with longitudinal, almost warty, stretch marks. *Boletellus minimatenebris* has some common morphological characters with *B. chrysenteroides*, but they differ because the latter has a larger pileus (56–80 mm), dry, areolate at maturity, and with a whitish context, and short basidiospores ranging from 9 to 14 (–17) × 6–8 μm, with deep and short longitudinal striations. In addition, *B*. *chrysenteroides* belongs to sect. *Chrysenteroidei* [39]. Phylogenetic analysis shows, with 98ML/1IB support, that *Boletellus minimatenebris* belongs to sect. *Boletellus*, with *Boletellus* sp. (HKAS122526) from China as the same species [76].

***Cacaoporus mexicanus*** Ayala-Vásquez, García-Jiménez, del Valle, **sp. nov.** (Figure 8D–F, Figure 9C,D and Figure 10).

MycoBank no: MB 834539.

Etymology: named *mexicanus* in honor of the country where the species is described.

Holotype: Mexico, Oaxaca State, Mixistlán de la Reforma municipality, Santa María Mixistlán Town, Yootsa’am Place, 25 July 2017, N 17°08′40.79″ W 96°05′20.04″, Ayala-Vásquez O. (ITCV-978, MEXU-HO-30128, RPB2 GenBank Num: OQ938898; ATP6 OR683446).

Diagnosis: basidiomata: small, dark brown to chocolate brown, context grayish brown, chocolate-brown reddening when bruised, hymenophore pale brown to brown; basidiospores: (7) 8–14 (15.2) × (3.8) 4–5 (6) μm; caulocystidia: cylindrical to subclavate, scattered with small brownish to cinnamon crystals on the walls.

Description: pileus: 11–40 mm in diameter, convex, dark brown (8D4–8E4), chocolate brown (8F6–8F5), dark violet (17F8), furfuraceous, minutely tomentose, slightly cracked at the top when mature, margin somewhat rolled to incurved at maturity. Hymenophore: adhered, pores of 0.1–1 mm in diameter, circular when young, becoming angular to irregular at maturity, pale brown (5E3) to brown (5F7); tubes: 3–6 mm long, pale brown to dark brown (8F8) in mature specimens, chocolate brown when young. Context: 5–8 mm thick, whitish to grayish brown (11F3), gray–brown (10D1–10F1), chocolate brown (5F5), slightly red–brown, and reddening (10E8–10F8) to vinaceous (10E8–10F8) when cut, especially on the base of the stipe. Stipe: 33–50 × 4–8 mm, cylindrical, grayish brown (11F3)—especially at the base, the middle and upper parts being chocolate brown (8D4–8E4)—dark brown (5F8), furfuraceous, minutely tomentose to scabrous, with basal whitish mycelium, turning reddish gray (11C6) to violet–brown (11D6) when touched. Odor: fungoid. Taste: pleasant. Chemical reaction: pileus turns red, hymenophore and context turn dark brown, and stipe turns red with NO_4_OH; all basidiomata turn blackish in KOH. Basidiospores: (7) 8–14 (–15.2) × (3.8–) 4–5 (–6) μm (Q = 1.96–2.46, Qm = 2.1, *n* = 175, from three basidiomata constituting the type, two paratypes), ovoid, amygdaliform with subacute apex in lateral view, brown in KOH, brown–orange in Melzer’s reagent, with suprahilar depression visible in some spores. Basidia: 30–37 (–42) × 7–8 (–9) μm, clavate, four-spored. Hymenophoral trama: divergent, *Boletus*-type, middle stratum of hyaline to brownish hyphae of 5–8 (–10) μm in diameter, cylindrical, thin-walled; lateral stratum with hyphae of 3–8 (–11) μm, some with gelatinized wall, thick-walled. Pleurocystidia: 34–37 (–42) × 4–7 (–10) μm, clavate, ventricose-rostrate, hyaline or brown in KOH, yellowish brown in Melzer’s reagent, thin-walled. Cheilocystidia 40–50 (–58) ×7–9 (–10) μm, ventricose-rostrate, clavate, obclavate, pale brown in KOH, yellowish brown in Melzer’s reagent, thin-walled. Pileipellis: composed of a trichoderm with terminal cells of 15–33 (–42) × 4–7 (–8) μm, cylindrical, subclavate, pale brown, thin-walled. Caulocystidia: 14–37 (-51) × 4–5 (–6) μm, cylindrical to subclavate, thin-walled, scattered with small brownish to cinnamon crystals on the walls in KOH or Melzer’s reagent; caulobasidia: 45–53 (–56) × 6–9 μm, cylindrical to subclavate hyaline or brown in KOH, yellowish brown to cinnamon in Melzer’s reagent, four-spored, with short sterigmata.

**Habitat and distribution**: solitary or growing scattered, associated with *Quercus scytophylla* and *Quercus* spp. forests from July to September at 2270 m.a.s.l. Currently identified in Oaxaca, Mexico.

**Additional specimens examined**: Mexico, Oaxaca state, Mixistlán de la Reforma Municipality, Santa María Mixistlán Town, Yootsa′am Place, 22 July 2017, N 17°08′39″ W 96°05′22″, Ayala-Vásquez, O. (ITCV-870, MEXU-HO-30122); Yootsa’am Place, *Quercus* forest, 25 July 2017, N 17°08′24.15″ W 96°05′17.03″, Ayala-Vásquez, O. (ITCV-903); Yootsa’am Place, 5 August 2017, 17°8ʹ24.15″N 96°05ʹ19″W, Ayala-Vásquez, O. (ITCV-904, MEXU-HO 30110, RPB2 GenBank No: OQ938897); Yootsa’am Place, *Quercus* forest, 03 September 2017, N 17°08′46.05″ W 96°05′18.33 ″, Ayala-Vásquez, O. (ITCV-1023, MEXU-HO 30103, RPB2 GenBank No: OQ938899); Yootsa’am place, 26 August 2017, N 17°08′40.62″ W 96°05′20.94″, Ayala-Vásquez, O. (ITCV-1024); and Yootsa’am place, 3 September 2017, N 17°08′40″ W 96°05′20.49″, Ayala-Vásquez, O. (ITCV-1027, RPB2 GenBank No: OQ938896. ATP6 OR683445).

**Notes**: *Cacaoporus* is a genus described from Thailand five years ago [63]. *Cacaoporus mexicanus* is a new species supported with morphological and phylogenetic data (100ML/1PPB). *Cacaoporus mexicanus* is a sister group of *Cacaoporus* sp. (SV0402) with support of 96ML/1BI; it is characterized by dark brown to chocolate basidiomata but it differs due to its pale brown hymenophore, longer basidiospores ((7–) 8–14 (–15.2) × (3.8–) 4–5 (–6) μm) and pleurocystidia (34–37 (-42) × 4–7 (–10) μm), and because it is clavate, ventricose-rostrate, and is found under *Quercus scytophylla*, which is endemic to Mexico and is currently only known in five states. *Cacaoporus pallidicarneus* differs due to its dark brown hymenophore, the shorter and narrower ((6.5–) 6.7–7.7–8.6 (–11.5) × (3.8–) 4–4.6–5.1 (–5.5) μm) basidiospores, the longer (44–) 44.2–54.7–67.6(–68) × (5–)5–6–7(–7) µm) pleurocystidia, and its discovery under Fagaceae trees (*Dipterocarpus* spp., *Shorea* spp., *Lithocarpus* sp., *Castanopsis* sp., and *Quercus* sp.), while *C. tenebrosus* has a hymenophore with longer basidia ((33.6–)34.3– 38.8–45.8(–47) × (7.7–)7.8–9.5–10.8(–10.9) µm) and scattered cystidia with small brownish-yellow to yellowish-brown crystals on the walls in KOH or NH_4_OH [66].

***Garcileccinum* gen. nov.** Ayala-Vásquez, Pérez-Moreno.

MycoBank no: 849921.

Etymology: named in honor of Dr. Jesús García-Jiménez, eminent Mexican mycologist and pioneer in the study of the Mexican boletes.

Diagnosis: basidiomata: pink salmon, grayish, pearl gray, grayish orange, then cinnamon brown to mustard brown to brown–orange color. Hymenophore: cream, yellowish white, grayish orange; context whitish, bruising pale grayish, vinaceous to dark violet, gray–violet, pale blue–green to greenish blue to deep blue at the context stipe base, and often slowly developing scattered orange–pink or coral pink stains; interior: white, changing to pinkish at apex, bluish green in the base. Stipe: finely floccose upper half, pruinose to scabrous, dry, white, apricot yellow, at first, becoming a pale caramel color, grayish orange. Basidiospores: smooth, fusoid to subfusoid. Pileipellis: composed of an ixotrichodermium embedded in a gelatinous matrix.

Type species: *Garcileccinum salmonicolor* Ayala-Vásquez, Pérez-Moreno, Pinzón.

***Garcileccinum salmonicolor*** Ayala-Vásquez, Pérez-Moreno, Pinzón, gen. nov. sp. nov. Figure 8G–I and Figure 11.

MycoBank no: MB 834539.

Etymology: The name is an allusion to the basidiomata color.

Holotype: Mexico, Oaxaca state, Mixistlán de la Reforma Municipality, Santa María Mixistlán Town, Kuyukeexp Place, 27 July 2017, N 17°07′21′′ W 96°05′50′′, O. Ayala-Vásquez (ITCV-914, MEXU-HO 30108, ITS GenBank Num: OQ971866, nrLSU GenBank Num: OQ909093, RPB2 GenBank Num: OQ938917, TEF1 GenBank Num: OR683442).

Diagnosis: basidiomata: pileate-stipitate, pink salmon, salmon, apricot yellow to gray reddish, context edge pink salmon to salmon. Basidiospores: (10–) 11.2–16.5 (–17.5) × 3.5–4.5 (–5.5) µm, subfusoid to fusoid.

Description: pileus: 44–46 mm in diameter, convex, somewhat dry to viscid when wet, subtomentose to subrugulose when young, areolate when mature, salmon-colored, apricot yellow (5B6–5B4), gray reddish (9A5–9A6) to pink salmon (10A2) when young, orange–brown (10A3) to gray–brown salmon (5B4) when mature, sterile margin somewhat appendiculate. Hymenophore: adhered; pores: 0.5–0.8 mm in diameter, circular to irregular, white yellowish (3A3) to apricot yellow (5B4–5B6) when young, orange–gray (5B3) when mature; tubes: 8 mm in length, concolorous to the pores, bruising pale brown (5C5) to brown–orange (5D5) when cut. Context: 6 mm thick, compact, cream color, salmon (10A2) mostly at the edges of the pileus and stipe context, stipe context whitish, pale yellow to salmon and green (28B5) to blue–gray (23C5, 22D7) at its base after being cut. Stipe: 48–50 × 8–10 mm, clavate, scabrous to pseudoscabrous, salmon (10A2), apricot yellow (5B6–5B4), pale red (10A3), pastel red (10A4), changing from brown (5E8) to cinnamon brown (9A5–9A6) when touched. White mycelium changes to salmon color when touched. Chemical reactions: pileus and hymenophore turn brown (6E8) in KOH without color change at the stipe and context, pileus turns brown in NH_4_OH and hymenophore turns reddish brown (9E8). Basidiospores: (10–) 11.2–16.5 × 3.5 –4.5 (–5.5) µm (Q = 2.21–2.6, Qm = 2.82, *n* = 35, two basidiomata), subfusoid to fusoid, yellow to yellowish brown in KOH, with suprahilar depression, thick-walled. Basidia: 20–30.5 × 8–12 um, clavate, broadly clavate, four-spored, hyaline in KOH, thin-walled. Pleurocystidia: 36–60 × 10–15 µm, fusoid-ventricose, hyaline in KOH, thick-walled. Cheilocystidia: 31–45 × 9–11.5 µm, fusoid-ventricose, hyaline in KOH, thick-walled. Hymenophoral trama: *Boletus*-type, middle stratum of 6–17 µm; lateral stratum of 3.5–11 µm, hyaline to pale yellow in KOH. Pileipellis: composed of an epithelium, with terminal cells of 21–62 × 5–6.5 µm, subelongate to cylindrical, hyaline in KOH, with rounded to obtuse apex, thick-walled. Stipitipellis: hyphae erect, parallel, giving rise to clusters of caulocystidia of 35–68 × 4–8 um, subfusoid, subelongate, hyaline in KOH, thin-walled.

**Habitat and distribution**: solitary, associated with *Quercus rugosa* in *Pinus*–*Quercus* (*Pinus teocote*, *P. oaxacanus*, and *Quercus rugosa*) and *Quercus–Pinus* forests (*Quercus rugosa*, *Quercus spp. Pinus teocote*, and *P. oaxacanus*) from July to October at 1530 to 1540 m.a.s.l. Currently identified only in Oaxaca, Mexico.

**Material examined:** Mexico, Oaxaca state, Santiago Zacatepec District, Mixistlán de la Reforma Municipality, Santa María Mixistlán Town, Nuuptëëkëpx place, N 17°07′ 27′′03 W 96°05′50′′, October 2017, Ayala-Vásquez O. (1070-ITCV, MEXU-HO 30105, ITS GenBank Num: OQ971867, nrLSU GenBank Num: OQ909094, RPB2 GenBank Num: OQ938918, TEF1 GenBank Num: OR683443).

**Notes**: *Garcileccinum salmonicolor* differs from other species in the genus due to its pileate-stipitate, pink salmon, salmon, apricot yellow to gray-reddish basidiomata, with pink salmon to salmon context edge; due to the surface of its pileus, that is subtomentose or subrugulose to light viscid when humid or whitish, pale yellow to salmon, and immutable when the pileus context is cut; due to the whitish and pale yellow to salmon stipe context; due to the green (28B5) to blue–gray (23C5, 22D7) base; and due to the length (10–) 11.2–16.5 × 3.5 –4.5 (–5.5) µm) of its basidiospores. Meanwhile, the pileus surface of *Garcileccinum viscosum* is always viscid, subrugulose to rugulose at first, and becomes reticulate-pitted, grayish orange, apricot yellow, cinnamon brown to mustard brown, and white changing to pinkish at its apex. The base stipe context is bluish green, and the basidiospores are 12.6–17.5 × 4.9–6.3 µm. *G. viscosum* occurs under *Quercus peduncularis* Née and *Pinus caribaea* Morelet, while *G. salmonicolor* is distributed in mixed forest putatively associated with *Quercus rugosa* at altitudes between 1530 and 1540 m.a.s.l. *Garcileccinum salmonicolor* differs from *Garcileccinum violaceotinctum* by its pearl gray to cinnamon brown color. When cut, it turns turquoise-tinted. Its basidiomata are pale grayish and vinaceous to dark violet. Its basidiospores are 12–13.6 (–16) × 4-4.8 (–5.6) µm. *G. viscosum* and *G. violaceotinctum* were originally described from Belize, found under *Pinus caribaea* and *Quercus* sp. [17]. In Mexico, *G. viscosum* is distributed in a *Pinus–Quercus* mixed forest occurring under *Pinus teocote*, *P. oocarpa*, and *Quercus rugosa*.

***Garcileccinum viscosum*** ((Halling and B. Ortiz) Mikšík) Ayala-Vásquez, Pérez-Moreno, **comb. nov.**

MycoBank no: MB 849,922.

Basionym: *Leccinum viscosum* Halling and B. Ortiz, in Ortiz-Santana and Halling, Brittonia 61(2): 172 (2009).

Synonym: *Leccinellum viscosum* (Halling and B. Ortiz) Mikšík, Index Fungorum 304: 1 (2016).

Typification: Belize. Cayo: Mountain Pine Ridge, entrance road to Five Sisters Lodge,

335 m, 6 Oct 2003, Halling 8528 (holotype: NY; isotypes: BRH, CFMR).

***Garcileccinum violaceotinctum*** (B. Ortiz and T.J. Baroni) Ayala-Vásquez, Pérez-Moreno **comb. nov**.

MycoBank no: MB 849923.

Basionym: *Leccinum violaceotinctum* B. Ortiz and T.J. Baroni, Fungal Diversity 27(2): 352 (2007).

Typification: Belize, Belize District, Belize Zoo area near Democracia, at the Tropical Education Center, N 17°21′27′′ W 88°32’30”, 30 m.a.s.l., 6 October 2002, BOS 327, BZ1676 (CMFR, holotype).

***Leccinum oaxacanum*** Ayala-Vásquez, Martínez-Reyes, and González-Martínez, **sp. nov**. (Figure 12 and Figure 13D–F).

MycoBank no: MB 849924.

Diagnosis: this species is characterized by its medium-sized basidiomata; brownish red, reddish brown to cinnamon pileus with tomentose surface; and whitish, changing to greenish blue, or grayish violet context. After it is cut, the context of the stipe turns bluish gray to greenish blue. Basidosphores are (10–) 11–14 (–15) × 4–5 µm, and the pileipellis is composed of a trichoderm with prostrate hyphae.

Holotype species: Mexico, Oaxaca State, Santiago Zacatepec District, Mixistlán de la Reforma Municipality, Santa María Mixistlán Town, 1 km, 29 June 2022, N 17°09′39′′ W 96°04′32′′, 2498 m.a.s.l., Ayala-Vásquez, O. (MEXU-HO 30460, nrLSU GenBank Num: OQ909096, RPB2 GenBank Num: OQ938915, TEF1 GenBank Num: OR683444).

Etymology: the name refers to the state where the species was identified.

Description: pileus: 28–38 mm, broad convex to convex, brownish red (9C8), reddish brown (8D8) to cinnamon, tomentose surface, entire margin sterile. Hymenophore: adhered, pores 0.1–0.3 mm and whitish, changing to grayish yellow (4C4–4C5) when touched, tubes: 5–7 mm long, whitish, turning greenish blue (24B8) slowly when cut and, later, olivaceous (4C5). Context: 11–13 mm thick, whitish when cut, greenish blue (24B8) and, after a few minutes, grayish violet (18E4); stipe context turns bluish gray (19E3,19E4) with some greenish blue (24B8) tones. Stipe: 35–50 × 18–20 mm, clavate with attenuated base, scabrous surface, white when young, cinnamon (8D8), reddish brown (8E8), whitish apex at maturity; when cut, turns gray–brown (8E3). Slightly sweet taste, fungoid smell. Basidiospores: (10–) 11–14 (–15) × 4–5 µm (Q = 2.5–2.9, Qm = 2.25, *n* = 35), fusoid to subfusiform, yellowish to yellowish brown in KOH, yellow to pale brown in Melzer’s reagent, smooth, thin wall. Basidia: 30–36 × 9–10 µm, clavate, four sterigmata, hyaline in KOH, with granular content visible in Melzer’s reagent, tetrasporic. Pleurocystidia: 34–36 × 9–10 µm, fusoid to ventricose-rostrate, hyaline to pale yellow in KOH, yellow in Melzer’s reagent. Cheilocystidia: 35–53 × 7–10 (–14) µm, fusoid, ventricose-rostrate to clavate, hyaline to pale yellow in KOH, yellow in Melzer’s reagent. Stipitipellis: composed of tubular to cylindrical hyphae of 300-400 µm, with caulobasidia of 28–33 × 8–10 µm, four sterigmata; caulocystidia: 35–60 × 8–10 µm, fusoid, ventricose-rostrate, thick-walled. Pileipellis: composed of a trichoderm with prostrate hyphae, with terminal hyphae (36–) 45–57(–60) × (5–) 6–10 µm, cinnamon to brown–orange in KOH, with visible content in Melzer’s solution.

**Material examined**: Mexico, Oaxaca state, Santiago Zacatepec District, Santa María Tlahuitoltepec Municipality, Santa María Yacochi Town, 0.5 km, 24 August 2023, N 17°09′32′′ W 96°03′59′′, 2729 m.a.s.l., Ayala-Vásquez, O., AVO-1177 (MEXU-HO 30705).

**Habitat and distribution:** solitary, distributed in disturbed cloud forest, associated putatively to *Arbutus Xalapensis* Kunth. Currently, it has been identified only in Mexico, Oaxaca state, Mixistlan town.

**Notes***: Leccinum oaxacanum* is phylogenetically found in the *Leccinum* s.s. clade with 100 ML/1IB support. It shares macromorphological characteristics, mainly the pileus color, with the rest of the species in the genus. However, *Leccinum oaxacanum* differs from other species due to its medium-sized basidiomata, tomentose pileus surface, whitish context that becomes greenish blue to grayish violet when cut, stipe context that turns bluish gray to greenish blue, and basidiospores that are (10−) 11–14 (–15) × 4–5 µm, 2–5 µm. Morphologically, this species is related to *L. manzanitae* (NY14041), described by Thiers [77] and associated with *Arbutus menziesii* Pursh and *Arctostaphyllus* spp. in the coastal areas of California, USA. *L. manzanitae* is characterized by its large basidiomata, dark red, viscid, reticulate, and occasionally tomentose pileus surface, white context that changes to fuscous (especially when young), and the fact that it is never reddish. In addition, it is characterized by subellipsoid, fusoid, subcylindrical to inequilateral basidiospores of 13–17 × 4–5.5 µm, a trichoderm pileipellis that is composed of free, tangled hyphal tips, and elongated terminal cells that are often tapered and possess ochraceous contents. Another related species is *L. monticola,* which differs due to its sterile flap margin pileus; white with brown to black scabrous stipe; context that is white, changing to fuscous and blue–green; basidiospores that are 15–18.9 × (3-) 4.9–5.6 µm; and its association with *Comarostaphylis arbutoides* Lindley in Costa Rica [31].

***Leccinum juarenzense*** Ayala-Vásquez and Pinzón, **sp. nov.** (Figure 13A–C and Figure 14).

MycoBank no: MB 849925.

Etymology: named after the Sierra Juarez region where the species was found.

Holotype: Mexico, Oaxaca State, Ixtlán de Juárez District, Santa Catarina Ixtepeji Municipality, La Cumbre Ixtepeji Town, km 2.3 towards the cabins, 7 October 2017, longitude N 17°19′51′′ W 96°62′71′′, Ayala-Vásquez (1117-ITCV, MEXU-HO 30112, holotype, nrLSU GenBank Num: OQ909095, RPB2 GenBank Num: OQ938916, GenBank: RPB1 Num: OQ971865).

Diagnosis: pileus: grayish magenta, purplish gray, grayish brown, rugulose, whitish hymenophore; context: whitish, turning pale red when cut; stipe context: brown vinaceous tones changing to turquoise blue at context base; basidiospores: (11−) 13–18 (−20) × 5–6 (−7) μm; pileipellis: composed of an epithelium.

Description: pileus: 40 mm in diameter, broadly convex to convex, grayish magenta (13E5, 13E4) to purplish gray (13E3) at the center, grayish brown (7D3, 7E3) at the edge, rugulose, somewhat moist, crenulate margin. Hymenophore: adhered, pores of 0.3–0.5 mm, whitish, turning reddish brown (8D5) when touched; tubes: 4 mm length, whitish, turning pale vinaceous (8D3) when cut. Context: 6 mm thick, whitish, turning pale red (8D5) when cut; stipe context: whitish, turning brown vinaceous (10A4) to turquoise blue (24A4–24A5) in some parts when cut. Stipe: 80 × 9–12 mm, clavate, whitish when young, grayish brown (7D2, 7F4) when mature; apex: whitish, changing to light turquoise (24A5) when touched, scabrous to longitudinally striate. Taste: acid. Odor: fungoid. Chemical reaction: in NH_4_OH, the pileus turns brown (5E8), the hymenophore and context turn reddish orange (18A8), and the stipe turns yellow (3A8). Basidiospores: (11−) 13–18 (−20) × 5–6 (−7) μm (Q = 2.4–2.7, Qm = 2.37, *n* = 40, one basidiomata holotype), smooth, elliptical to fusiform, isodiametrical, green in KOH, brown in Melzer’s reagent, dextrinoid, with or without visible suprahilar depression. Basidia: 31–40 ×10–15 μm, clavate, four-spored, hyaline in KOH, brown in Melzer’s reagent. Pleurocystidia: 32–40 × 8–9 μm, fusoid, hyaline in KOH, yellow in Melzer’s reagent, thick-walled. Cheilocystidia: 46–65 × 10–12 μm, fusoid, ventricose-rostrate, with long apex, brown in KOH, brown–orange in Melzer’s reagent, with intracellular content, thick-walled. Pileipellis: composed of an ephitelium containing hyphae that are globose, subglobose, cylindrical, clavate, and, in some cases, isodiametric. Pileipellis composed of 19–27 × 6.5–13 μm, hyaline in KOH, yellowish in Melzer’s solution, and thick-walled hyphae. Caulocystidia: 30–45 × 10–16 μm, clavate, sometimes fusoid, hyaline in KOH, yellow in Melzer’s reagent, with a basal septum, thick-walled.

**Habitat and distribution**: solitary, occurring in *Quercus* sp. and *Quercus–Pinus* forests, in October, at 3000 m.a.s.l. Currently only identified in Oaxaca, Mexico.

**Notes**: *Leccinum juarenzense* is characterized by its very rugulose pileus (which is grayish magenta to grayish brown and purplish gray at the center), its whitish hymenophore (when touched, it turns vinaceous), its whitish context (when cut, it changes to pale red to brown vinaceous), and its turquoise base stipe. In addition, its basidiospores are (11-) 13–18 (20) × 5–6 (-7) μm and elliptical to fusiform, and its pileipellis is composed of an epithelium containing hyphae that are globose, subglobose, cylindrical, clavate, and, in some cases, isodiametric. Phylogenetically, *L. juarenzense* is the sister to *L. talamancae* described from a *Quercus* forest, from Costa Rica by [28]. *L. juarenzense* differs due to its brownish gray and cocoa brown to dark reddish brown pileus; its tomentose, pitted, and rugose pileus surface when young; its areolate pileus surface when mature; its white context (which changes slowly to pink to reddish orange); its dark blue base stipe; its basidiospores of 17.5–22.4 × 4.9–6.3 µm (these are subfusoid to ellipsoid); and its pileipellis, which is composed of an epithelium containing subisodiametric to spherical cells (in rare cases, these are subcylindric) and hyaline to brown pigment. *L. juarenzense* can be confused with *Leccinellum quercophilum* described by [78] from Illinois, USA, under *Quercus alba*, due to the similar colors of the basidiomata. However, *Leccinellum quercophilum* can be distinguished by its dry to glabrous pileus surface when young; its conspicuously areolate pileus surface at maturity; its bluish green color when touched; its creamy whitish, grayish brown, and brown hymenophore (when touched, it changes to a greenish color); its large basidiospores of 15–18(–28) × 5–7.5 µm; its fusiform character; and its pileipellis composed of an epithelium with terminal elements that are subglobose to clavate or irregular.

*Leccinum juarenzense* does not fit the generic diagnosis of *Leccinum* because of its change in coloration upon cutting and phylogenetic analysis. We agree with the work of Kuo et al. [16], who proposed that *L. talamancae* should be included in another genus: our phylogram shows that *L. juarenzense* belongs to the same genus. However, further studies are needed to propose the genus. *L. juarenzense* is a rare species, i.e., we only found one specimen during five years of sampling in the same region.

***Pulchroboletus neoregius*** (Halling and G.M. Muell.) Ayala-Vásquez; Pérez-Moreno and Martínez-Reyes, **comb.nov.** (Figure 13G–I and Figure 15).

MycoBank no: MB 849927.

Basionym: *Boletus neoregius* Halling and G.M. Muell., Mycologia 91(5): 897 (1999).

Description: pileus: 33–78 mm, broadly convex to convex when young, plane–convex at maturity, yellowish brown (5D4, 5E4), light brown (5D5, 5E4) when young, pale orange (5A4) to red–brownish (5C3), cinnamon to brown (6D7–6D5) in the center of the pileus, tomentose surface when young, subtomentose to smooth when mature, entire margin, adnexed to depressed around the stipe. Stipe: the pores are closed when young, 0.5–1 mm at maturity, circular, vivid yellow (3A8) to chrome yellow (5A7), brownish yellow (5C7) when cut (changing to pale green (26A4) when young), greenish blue (24A8–24A7) at maturity; tubes: 3-13 mm in length, pale yellow (changing to pale green (26A4) when they are cut while young), deep blue (21E8) at maturity. Context: 14–15 mm, spongy, pale yellow (30A4) to whitish (when cut, changing to pale green (26A4) to greenish blue (24A8–24A7)); context of stipe: pale yellow (30A4) with red–brown (8D7–8E7) tones at base and middle; when cut, changing to pale green (26A4) to greenish blue (24A8–24A7). Stipe: 82 × 15–25 mm, clavate, attenuated base, smooth surface with rivulose appearance, vivid yellow (3A8) when young, red–brown (8D7-8E7) at maturity; basal mycelium: vivid yellow (3A8). Odor and taste: sweet citrus. Basidiospores: (12–) 13–16 (–17) × 4–5μm (Q = 2.03–2.9, Qm = 2.27, *n* = 64, two basidiomata), cylindrical, subfusoid, with or without suprahilar depression, yellow in KOH, yellow–brown in Melzer’s reagent, anamyloid. Basidia: 35–43 × 10–12 μm, clavate, two to four sterigmata, hyaline to pale yellow in KOH, yellow in Melzer’s reagent. Pleurocystidia: rarely present, 60 × 10–7 μm, fusoid, hyaline in KOH, yellow in Melzer’s reagent, thin wall. Cheilocystidia: 27–48 × (7−) 8–12 μm, subclavate to subcylindrical (sometimes subfusoid with a septum in the third part of the structure), hyaline to pale yellow in KOH, yellow–brown in Melzer’s reagent, thin wall. Hymenophoral trama: boletoid, central elements of 3–5 μm, cylindrical, lateral elements 4–12 µm wide, divergent, hyaline to pale yellow. Caulocystidia: 22–38 × (6−) 7–10 μm, one to two septa, cylindrical to subclavate, thin wall, hyaline in KOH, yellow–brown in Melzer’s reagent. Caulobasidia: 20–30 × 8–10 μm, two to four sterigmata, hyaline in KOH, yellow–brown in Melzer’s reagent. Pileipellis: composed of a trichoderm 300–400 μm thick containing disorganized hyphae with terminal cells of 24–48 × 7–10 (11) μm, cylindrical, with rounded or sharp apex, hyaline to yellow in KOH, yellow–brown in Melzer’s reagent, thin wall.

**Habitat and distribution**: solitary; associated with *Quercus laurina*, *Quercus* sp., and *Abies religiosa*; identified in mixed *Pinus–Quercus* and mixed coniferous *Abies* forests from July to September at 2800 to 3286 m.a.s.l. Currently identified in Tlaxcala, Mexico.

**Additional Specimens Examined:** Mexico, Tlaxcala State, Nanacamilpa Municipality, San Felipe Hidalgo Town, Piedra Canteada private forest land, La Vaquería Place, 2800 m.a.s.l., 14 July 2021, Taboada-García, A.; Ayala-Vásquez O. (MEXU-HO 30423, nrLSU GenBank Num: OQ940034); 1 km from entrance to the property, 4 August 2021, Pérez-Moreno J., Ayala-Vásquez O., Martínez-Reyes M. (MEXU-HO 30424, ITS GenBank Num: OQ940041); La Curva Place, 4 August 2021, 3152 m.a.s.l., Ayala-Vásquez O. (MEXU-HO 30425, ITS GenBank Num: OQ940040, nrLSU GenBank Num: OQ940035); El Plano Place, 29 September 2021, 3286 m.a.s.l., Ayala-Vásquez O. (MEXU-HO 30426, ITS GenBank Num: OQ940039, nrLSU GenBank Num: OQ940033); and State of Mexico, Ocuilan Municipality, San Juan Atzingo, mixed forest, 1 September 2022, Martinez-Reyes M, Ayala-Vásquez O., 1183 (MEXU OH 30711).

**Notes:** the holotype species of *Pulchroboletus neoregius* was described from Costa Rica by Halling et al. [30] as *Boletus neoregius.* The revised specimens differ in the color of the pileus, which is brown and, when touched, becomes dark red to red–brown, while the type specimen described by Halling et al. [30] has a deep red or pink to brown pileus; phylogenetic analyses show that the species belongs to the genus *Pulchroboletus* with a support of 100 ML/1BP. *P. neoregius* is used as a food resource by the Tlahuica-Pjiekakjoo culture. However, this knowledge is at risk of disappearing because only one family in the sampling locality regularly and safely consumes it.

*Tylopilus leucomycelinus* (Singer and M.H. Ivory) R. Flores and Simonini, *Journal of Mycology* 43 (2): 132, 2000 (Figure 16H,I).

**Notes:** most macroscopic and microscopic features of Mexican specimens are similar to those of previously reported specimens from Central America and the Dominican Republic [26,35]. However, it is interesting to note that the basidiomata examined in the present work had a bitter taste, in agreement with Singer et al. [79], differing from the mild taste reported by Gelardi et al. [26]. The size of the cystidia, especially the cheilocystidia, is similar to that described by the former authors but are shorter than those cited by the latter (25–49 × 6–13 (–17) µm vs. (38) 42–75 (83) × 7–16 (19) μm, respectively). The pores in Mexican specimens are stained brown, a character that was also recorded in specimens from the Dominican Republic by Gelardi [26], while those from Honduras were described as unstaining [79], they remained unchanged. In the specimens studied, we observed that most of the pileipellis elements were configured in an intermixed arrangement, but in some areas of the pileus, especially towards the surface, they were similar to a cutis; thus, the variation observed in Mexican specimens includes features observed previously, “a trichoderm of intricately interwoven hyphae” [26] and “a cutis consisting of strongly interwoven to occasionally erect hyphae” [26]. *Tylopilus leucomycelinus* was recorded previously in Central America and some Caribbean countries (including Belize, the Dominican Republic, Guatemala, and Honduras) [26,80], and now, its geographic extension is broad and ranges to eastern Mexico. In addition to having been previously known from the *Pinus* species (*P. oocarpa, P. caribaea,* and *P. occidentalis*), the Mexican record demonstrates its association with *Quercus sapotifolia* and *Q. oleoides* in tropical *Quercus* forests.

**Habitat and distribution**: identified in central America and the Caribbean. In Mexico, it was found growing in monodominant areas of *Q. sapotifolia* and monodominant areas of *Q. oleoides* in tropical *Quercus* forests.

**Additional Specimens Examined:** Mexico, Veracruz State, Zentla Town, 6 July 2016, Hervert 111, Montoya 5274; 12 July 2016, JC Corona 1298; August 10, 2016, Montoya 5279; June 22, 2017, Montoya 5313; July 12, 2017, Montoya 5241, Montoya 5242, Montoya 5243; Sep 14, 2017, M. Caballero 53 (all at XAL).

***Tylopilus pseudoleucomycelinus*** Ayala-Vásquez, Pinzón and Montoya, **sp. nov.** (Figure 16A–C and Figure 17).

MycoBank no: MB 849928.

Etymology: from the Greek “pseudo”, meaning similarity or false, in reference to *T. leucomycelinus.*

Holotype: Mexico, Oaxaca State, Mixistlán de la Reforma Municipality, Santa María Mixistlán Town, Kuuyu’ukeexp Place, N 17°07′32′′ W 96°05′38′′, 03 October 2017, Ayala-Vásquez (ITCV 1074, MEXU-HO:30115 duplicate, ITS GenBank Num: OQ940043, nrLSU GenBank Num: OQ940037, RPB2 GenBank Num: OQ938903).

Diagnosis: basidiomata: small. Pileus: dry, furfuraceous. Hymenophore: adhered, white, changing from yellowish brown to brown when cut or touched. Stipe surface: furfuraceous, basidiospores of 5–7 × 4–5 µm, ovoid to lacrymoid. Pileipellis: composed of a trichoderm.

Description: pileus: 19–28 (40) mm in diameter, convex, dark yellow (4C8) to orange (7A3) when young, cinnamon and orange–brown (7C8) to brown (7D8–7E8) when mature, furfuraceous, dry, incurvate margin. Hymenophore: adhered, with pores of 0.25–0.5 mm in diameter, circular, whitish. Tubes: 3 mm long, concolorous to the pores, turning brown–orange (5E7) when touched or cut, especially when mature. Context: 4 mm thickness, whitish, unchanged when cut. Stipe: 13–15 (−25) × 5–6 (−18) mm, subcylindrical to clavate, yellow (4A6), whitish near the apex, furfuraceous. Whitish mycelium present at the base of the stipe. Odor: garlic-like. Taste: astringent, minty after a few minutes. Chemical reaction: the pileus, context, hymenophore, and stipe turn brown (6F8) and pale brown (5F3), respectively, with NH4OH. The pileus turns brown–orange (5D8), the hymenophore and context turn pale brown (5D3), and the stipe turns grayish brown (6E3) in KOH. Basidiospores: 5–7 (–10) × 4–5 µm (Q = 1.25–1.42, Qm = 1.29, *n* = 105), ovoid, lacrymoid to ellipsoid, smooth, olivaceous in KOH, amyloid in Melzer’s reagent, thin-walled. Basidia: 28–30 (−45) × 8–9 (−9.6) μm, clavate, hyaline in KOH, two to four spores, thin-walled. Pseudocystidia: 30–35 (–66) × 7–8 μm, fusoid, fusoid-ventricose, brownish with intracellular content in Melzer’s reagent, thick-walled. Cheilocystidia: 33–66 × 7.2–12 μm, fusoid-ventricose, brownish, with intracellular content in Melzer’s reagent, thick-walled. Hymenophoral trama: bilateral, divergent, *Boletus* holotype; central stratum: compact, subregular, hyphae of 2–8 µm in diameter, cylindrical, pale yellowish to brown, some with brown–yellowish dense contents, thin-walled; lateral stratum: indistinctly gelatinous, with hyphae of 4–9 (–14) µm in diameter, cylindrical to broad, hyaline to pale yellow, thick-walled. Pileipellis: 150–200 μm thick, composed of a trichoderm of intertwingled hyphae, terminal cells of 30–40 × 7–9 μm, cylindrical, hyaline to pale yellow in KOH, yellow in Melzer’s reagent. Caulocystidia: 25–60 × 7–11 μm, fusoid, ventricose-rostrate to lanceolate, hyaline, with yellow intracellular content in KOH, yellowish brown with cinnamon intracellular content in Melzer’s reagent, thick-walled.

**Habitat and distribution**: solitary to scattered, growing under *Q. rugosa* and *Q. scytophylla* in *Pinus–Quercus* and *Quercus–Pinus* forests from September to October, at 1737–1926 m.a.s.l. To date, only identified in Oaxaca, Mexico.

**Additional Specimens Examined:** Mexico, Oaxaca State, Mixistlán de la Reforma Municipality, Santa María Mixistlán Town, Moontxa’am Place, Ayala-Vásquez, 14 September 2017 (1039-ITCV), Kuuyu’ukeexp Place, 24 August 2023, Ayala-Vásquez O., 1181 (MEXU-HO-30709); Moontxa’am Place, Ayala-Vásquez, 24 August 2023 (MEXU HO 30708).

**Notes**: *Tylopilus pseudoleucomycelinus* has small basidiomata, an orange pileus when young (which changes to orange–brown when it matures), and an attached hymenophore with whitish pores and tubes, which turn yellow–brown to brown when they are touched or cut. Its cystidia are larger than those of *T. leucomycelinus* [26] (Table 7). Phylogenetic analysis shows that it belongs to the *T. balloui* complex clade and is a sister to *T. leucomycelinus*. The specimen, previously named *Tylopilus balloui* (FMNH 1073250), recorded from Tamaulipas, Mexico, is not described in detail, although it is mentioned by Halling et al. [24]. It belongs to *T. pseudoleucomycelinus.*

***Xerocomus hygrophanus*** Ayala-Vásquez, Pinzón and García, **sp.nov.** (Figure 9E,F, Figure 16D–G and Figure 18).

MycoBank no: MB 849929.

Etymology: the Latin “*hygrophanus*” refers to the hygrophanous pileus surface, especially when young, which causes the pileipellis to become more transparent when wet and opaque when dry.

Holotype: Mexico, Oaxaca State, Santiago Zacatepec Mixe District, Mixistlàn de la Reforma Municipality, Santa María Mixistlán Town, Kuuyu’ukeexp Place, 3 October 2017, Ayala-Vásquez (ITCV-1073, MEXU-HO-30131, nrLSU GenBank Num: OQ975751, RPB2 GenBank Num: OQ938901).

Diagnosis: pileus surface: smooth to hygrophanous; stipe surface: smooth to hygrophanous, whitish; pileus context: whitish (changing to pale green when cut); basidiospores: (8−) 9−13 (−15) × 3−5 μm, elliptical to subfusiform, smooth under light microscopy, pinpricks under scanning electron microscopy.

Description: pileus: 16–38 mm in diameter, plane–convex, citrus yellow (3A8), wet surface, hygrophanous when young, pale brown, ferrugineus to brown at maturity with straight to incurved margins. Hymenophore: adhered, pores of 1–2 mm in diameter, angular, yellow (3A8) when young, pale brown at maturity; tubes: 3–6 mm long, concolorous to the pores, unchanging when young, pale green at maturity. Context: 2–6 mm thick, whitish to pale yellow when cut, changing to pale green (28A3) at maturity; context of stipe: white to brown at base. Stipe: 25−30 × 6−7 mm, cylindrical to subclaviform, smooth to fibrillose at maturity, whitish to pale yellow when young, yellow and pale brown to brown at maturity. Basal mycelium: whitish. Odor: fungoid. Taste: pleasant. Chemical reaction: pileus, hymenium, and stipe turn pale orange (6A3), dark brown (6F8), and orange–brown (5B4), respectively, in KOH. The pileus and hymenophore turn brown in NH_4_OH; the context and stipe do not change color. Basidiospores: (8-) 9−13 (−15) × 3−5 μm (Q = 2.18–2.48, Qm = 2.22, *n* = 105, basidiomata, holotype), elliptical to subfusiform, smooth under light optic microscopy, “pinpricks” under scanning electron microscopy, yellowish in KOH, brown in Melzer’s reagent. Basidia: 27−35 (−40) × 7−9 (−10) μm, claviform, four-spored, hyaline in KOH, pale brown in Melzer’s reagent. Pleurocystidia: 43–45 (−55) × 7−10 (11) μm, ventricose-rostrate, hyaline in KOH, thick-walled (1.5 μm). Cheilocystidia: 30−50 (−62) × 9−12 (−13) μm, napiform, fusoid to ventricose-rostrate, hyaline in KOH, pale brown in Melzer’s reagent, thick-walled. Hymenophoral trama: composed of subparallel, 5−14 μm, cylindrical hyphae. Pileipellis: composed of a trichoderm of 21−40 (−50) × 7−10 (−13) μm, clavate to tibiform (in some cases, septate), hyaline to yellowish in KOH, thin-walled terminal cells. Caulocystidia: 20−24 × 9−12 μm, clavate, hyaline to yellowish in KOH, thin-walled.

**Habitat and distribution**: scattered, growing associated with *Q. rugosa* in *Quercus–Pinus* forest. Currently only identified in Oaxaca, Mexico.

**Additional Specimens Examined:** Mexico, Oaxaca State, Santiago Zacatepec Mixe District, Mixistlán de la Reforma Municipality, Santa María Mixistlán Town, Kuuyu’ukeexp Place, 26 August 2023, Ayala-Vásquez O., AVO-1176 (MEXU-HO 30704).

**Notes**: the phylogram (ITS-nrLSU-RPB2) shows that *Xerocomus hygropanus* is a new species whose sequences nested with a sequence of an unidentified *Xerocomus* sp. (MAN061) from Costa Rica [35]. *Xerocomus illudens* is a sister species of *X. hygrophanus* with support of 0.96 BI/70ML. Additionally, the sequence named *Xerocomus illudens* (DD9854) described from North Carolina and Virginia, USA, is separated from the clade of *X. illudens* s.s., which indicates that *X. illudens* is probably a species complex [82], which requires further study. Morphologically, *X. illudens* and *X. hygrophanus* have similarities in their pale yellow coloration, but *X. hygrophanus* differs in the size of its basidiomata; its pale yellow to olive yellow hymenophore (which does not change color when cut); its white to pale yellow color (which changes to pale green when it is cut); its pileus context of (8−) 9−13 (15) × 3−5μm; its elliptical to subfusiform basidiospores; and its ornamentation, which is barely visible on scanning electron microscopy. On the other hand, *X. illudens* has a coarsely reticulate stipe, extending from two-thirds to its entire length, of 10–12 (16) × 4–5 (6) μm, with ellipsoid to subfusiform basidiospores [83].

## 4. Discussion

In the last five years, a number of new species of Boletaceae from Mexico have been described, demonstrating the great and unexplored diversity of this fungal group in the country [2,39,40,48,52,84]. Additionally, recently new genera have been described worldwide [13,27,34], for example, *Cacaoporus* [66], and others remain obscure, including genera belonging to the subfamily Leccinoideae [14,15].

The subfamily Leccinoideae currently includes hypogeous genera, such as *Octaviania, Chamonixia*, *Rossbeevera*, and *Turmalinea*; epigeous genera, such as *Kaziboletus*, *Leccinum* s.s. *Leccinellum* s.s., and *Spongispora*; and undefined *Leccinum* clades [12,85]. On the basis of morphological, molecular, and phylogenetic analyses, *Garcileccinum* is proposed as a new genus in the subfamily *Leccinoideae. Garcileccinum salmonicolor* gen. nov. sp. nov., its type species, is distributed in Oaxaca, Mexico, occurring under *Quercus rugosa.* Meanwhile, *G. viscosum* and *G. violaceotinctum* have been recorded in Belize [17,18]. Therefore, this genus is currently known only from Mesoamerica. *Leccinum oaxacanum* is found in the clade of *Leccinum* s.s. while *Leccinum juarenzense* is found in an undefined clade with *L. talamancae* Halling, L.D. Gómez and Lannoy. Our multi-locus analysis shows that *Leccinum* and *Leccinellum* are polyphyletic, as mentioned in previous works [15,16], and our phylogram shows undefined clades for some American sequences.

*Pulchroboletus* is a small genus, which currently includes only four species: *P. begoniinus* N.K. Zeng, Chang Xu and Zhi Q. Liang, described from China; *P. roseoalbidus* (Alessio and Littini) Gelardi, Vizzini, and Simonini, recorded in Italy; and *P. rubricitrinus* (Murrill) Farid and A.R. Franck and *P. sclerotiorum* M.E. Sm., Bessette and A.R. Bessette, described from the USA [86,87]. In this work, we added *P. neoregius*, distributed in Costa Rica and Mexico, as a fifth species in the genus.

*Tylopilus balloui* (Peck) Singer is a complex of species with the holotype being described from New York, USA. It has also been recorded in Australia, Mexico, Thailand [24], India [86], and China [9]. However, its phylogenetic analysis shows the complexity of the species since it has no basal support [9]. Halling [24] mentioned that *T. balloui* is actually a polymorphic species and that a more detailed study is needed to determine if it is a cryptic group of related species. Several authors have mentioned that obligate ectomycorrhizal fungi such as the genus *Tylopilus* migrated along with their hosts [24,88]; therefore, the origins of Pinaceae [89] and Fagaceae [90] in Mexico share similarities to those of the Asian species. Halling [24] carried out a distribution analysis of the *T.* complex *balloui* and considered that Mexico shares more similarity with Asian, Australian, and Central American specimens than with those from the USA. Gelardi et al. [26] mentioned that several authors consider that *T. balloui* s.s. is distributed in the eastern USA and Mexico and that the sequences reported from Central America belong to *T. leucomycelinus*. In our work, we consider that *T. leucomycelinus* is distributed in the Dominican Republic, Belize, and Mexico [1,26,81]. In Mexico, it is distributed in tropical *Quercus* forests, including *Q. oleoides* and *Q. sapotifolia*, while in Central America, it occurs in *Pinus* forests (including *P. oocarpa, P. caribaea*, and *P. occidentalis*) and, possibly, in *Quercus* forests. Mexico has a great diversity of *Quercus*, which includes more than 160 species, indicating, therefore, a great diversity of taxa within the *Tylopilus balloui* complex, which requires further study. *Tylopilus pseudoleucomycelinus* is distributed in *Pinus–Quercus* forests, forming a putatively ectomycorrhizal association with *Pinus teocote* and *P. oaxacanus* at 1737 to 1926 m.a.s.l. The hypothesis is that it is distributed from the Sierra Madre Oriental to the Coastal Plain of the Gulf of Mexico, according to the biogeographical classification proposed by Morrone [1,91].

*Xerocomus* s.l. has few sequences compared to the listed species in Index Fungorum [85]. The genus requires extensive study to resolve the identity of many species [35]. Our concatenated phylogram of nrLSU, ITS, and RPB2 (Figure 6) shows that *X. hygrophorus* belongs to the section *Xerocomus* s.s. [82]. *Xerocomus illudens* is a sister species described from the USA [83] and Mexico. In Mexico, it possesses biocultural importance for the Otomi-Hñähñu native culture from Queretaro state, where it is referred to as the Ixka hyethe (yellow or sulfureous) mushroom [92]. *Xerocomus hygrophanus* has been recorded from Mexico and Costa Rica (MAN2011-b-MAN061) [35]. In Mexico, it is putatively associated with *Quercus rugosa* in mixed forests. Previously, only *Xerocomus illudens* and *X. tenax* [2] were described as from Mexico, and in this work, we added *X. hygrophorus.*

Boletaceae is a family with great social, economic, and biocultural importance in Mexico, which is the country harboring the second-greatest number of edible mushrooms, just after China [8,9,10]. There are 86 edible wild species belonging to Boletales, which are consumed by members of the native ethnic groups in Mexico. In recent years, several new edible species have also been described, e.g., *Xerocomellus perezmorenoi* Ayala-Vasquez and M. Martinez-Reyes [48] and *Xerocomellus piedracanteadensis* Ayala-Vasquez, Perez-Moreno J., and Martínez-Reyes M. [52]. In this work, we record, for the first time, the edibility of *Pulchroboletus neoregius*, which is consumed by the Tlahuica-Pjiekakjoo culture who inhabit the state of Mexico; this culture consumes approximately 200 species of edible wild mushrooms. Additionally, it is important to continue researching species within the Boletaceae family due to the fact that some of them are only distributed in specific vegetation types, such as the cloudy forest, in danger of extinction due to great changes in land use.

## Figures and Tables

**Figure 1 jof-09-01126-f001:**
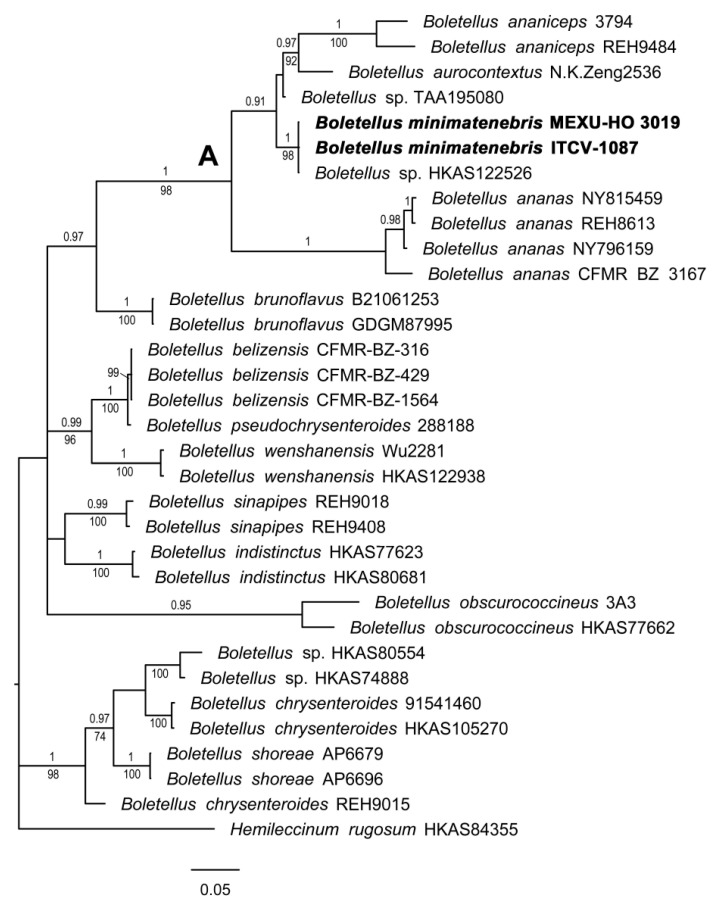
Phylogram of genus *Boletellus* with gene ITS, nrLSU, and RPB2. Maximum likelihood (ML)/Bayesian inference (BI) analyses were performed; the phylogram presented is the result of Bayesian inference. The bootstrap values (≥50%) and posterior probabilities (BI ≥ 0.90) are shown at the supported branches. *Hemileccinum rugosum* (HKAS84355) was used as the outgroup. The cluster where the new species is included is marked with the letter A. Newly generated sequences are indicated in bold.

**Figure 2 jof-09-01126-f002:**
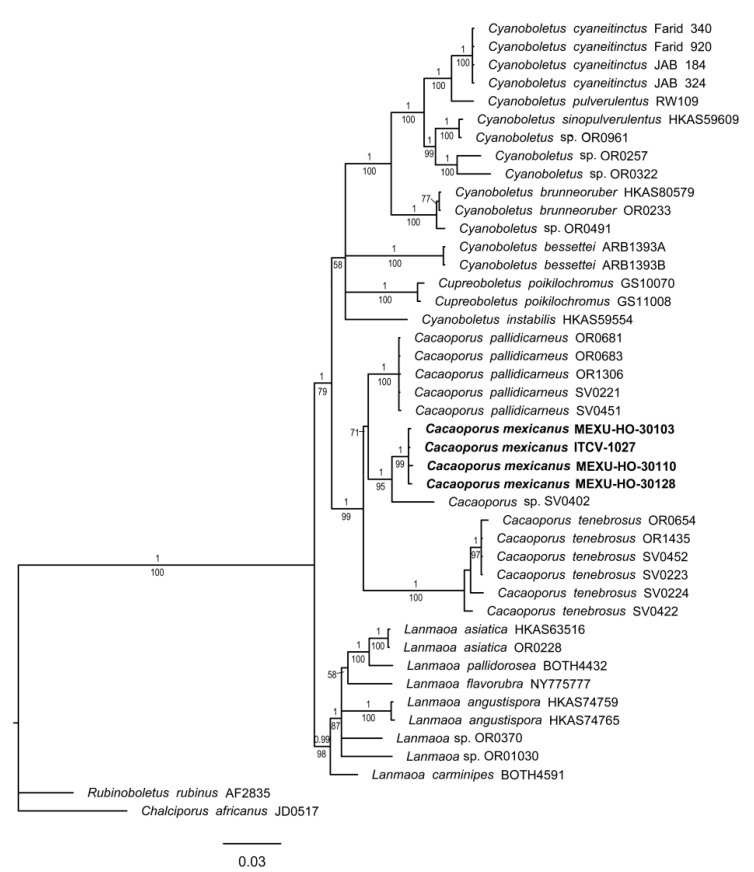
Phylogram of genus *Cacaoporus* with RPB2 and ATP6. Maximum likelihood (ML)/Bayesian Inference (BI) analyses were performed; the phylogram presented is the result of Bayesian inference. The bootstrap values (≥50%) and posterior probabilities (BI ≥ 0.90) are shown at the supported branches. *Cyanoboletus*, *Chalciporus africanus, Lanmaoa,* and *Rubinoboletus rubinus* were used as the outgroup. Newly generated sequences are indicated in bold.

**Figure 3 jof-09-01126-f003:**
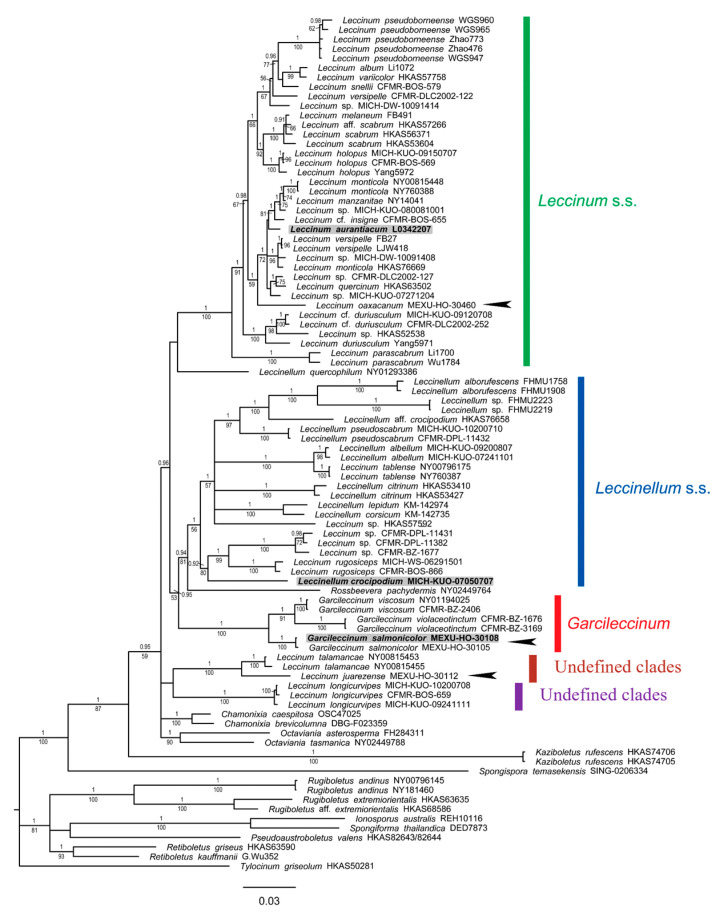
Phylogram of the subfamily Leccinoideae with nrLSU, RPB2, and TEF1. Maximum likelihood (ML)/Bayesian inference (BI) analyses were performed; the phylogram presented is the result of Bayesian inference. The bootstrap values (≥50%) and posterior probabilities (BI ≥ 0.90) are shown at the supported branches. *Rugiboletus*, *Ionosporus*, *Spongiforma*, *Pseudoaustroboletus*, *Retiboletus*, and *Tylocinum* were used as the outgroup; *Leccinum* genus *s.s.* (green color), *Leccinellum s.s.* (blue color), *Garcileccinum* (red color), undefined clades (wine color), undefined clades (violet color). Type specimens are highlighted with gray color, including *Garcileccinum salmonicolor* gen. nov. sp. nov., the type species of *Garcileccinum* gen. nov.

**Figure 4 jof-09-01126-f004:**
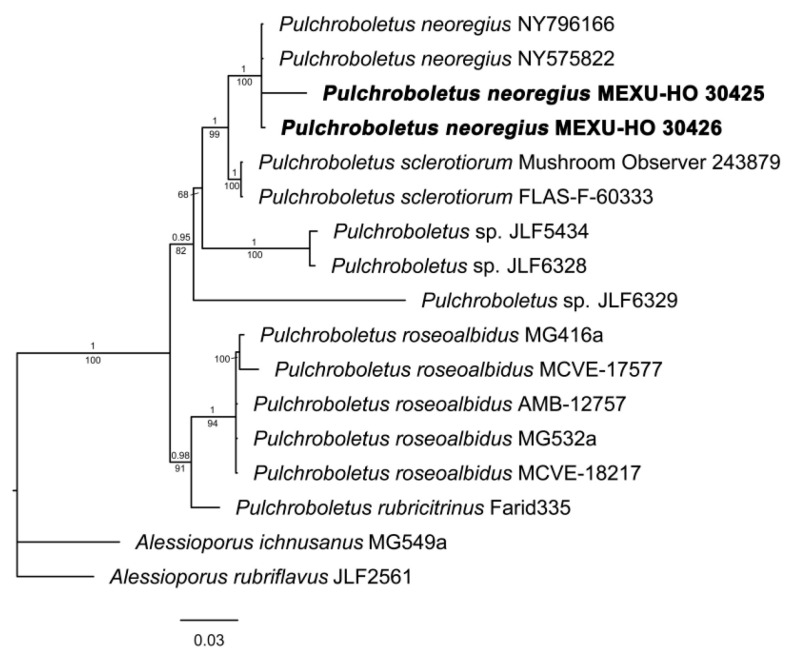
Phylogram of *Pulchroboletus* genus with gene ITS and nrLSU. Maximum likelihood (ML)/Bayesian inference (BI) analyses were performed; the phylogram presented is the result of Bayesian inference. The bootstrap values (≥50%) and posterior probabilities (BI ≥ 0.90) are shown at the supported branches. *Alessioporus ichnusanus* (MG549a) and *A. rubriflavus* (JLF2561) were used as the outgroup. Newly generated sequences are indicated in bold.

**Figure 5 jof-09-01126-f005:**
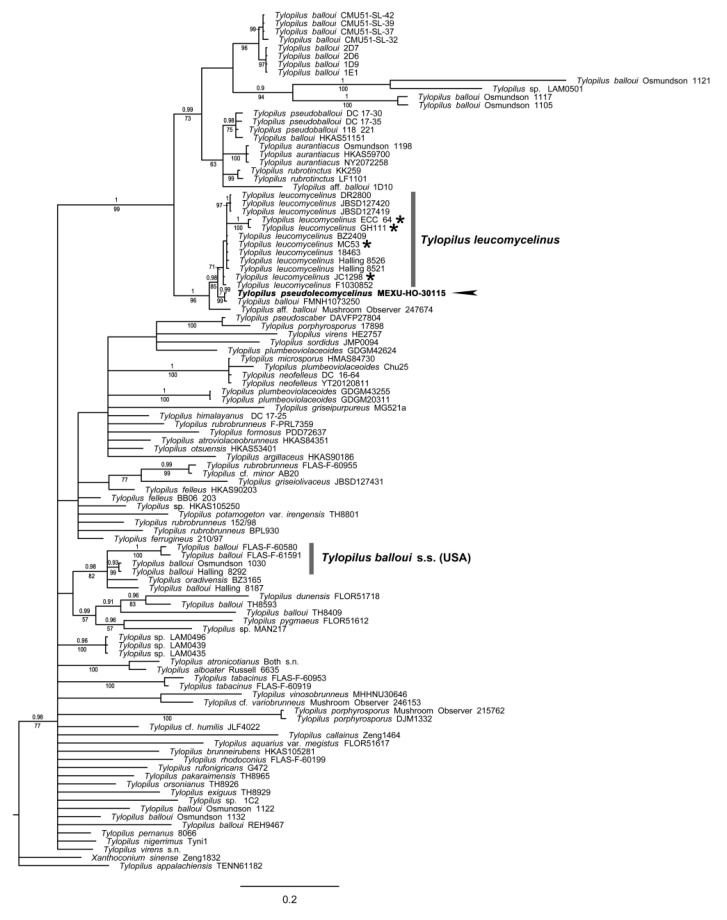
Phylogram of *Tylopilus* with ITS and nrLSU. Maximum likelihood (ML)/Bayesian inference (BI) analyses were performed. The phylogram presents Bayesian inference, bootstrap values (≥50%), and posterior probabilities (BI ≥ 0.90) at the supported branches. *Xanthoconium* was used as the outgroup. Newly generated sequences are indicated in bold. The symbol * shows sequences of *T. leucomycelinus* generated in this work as a new record for Mexico.

**Figure 6 jof-09-01126-f006:**
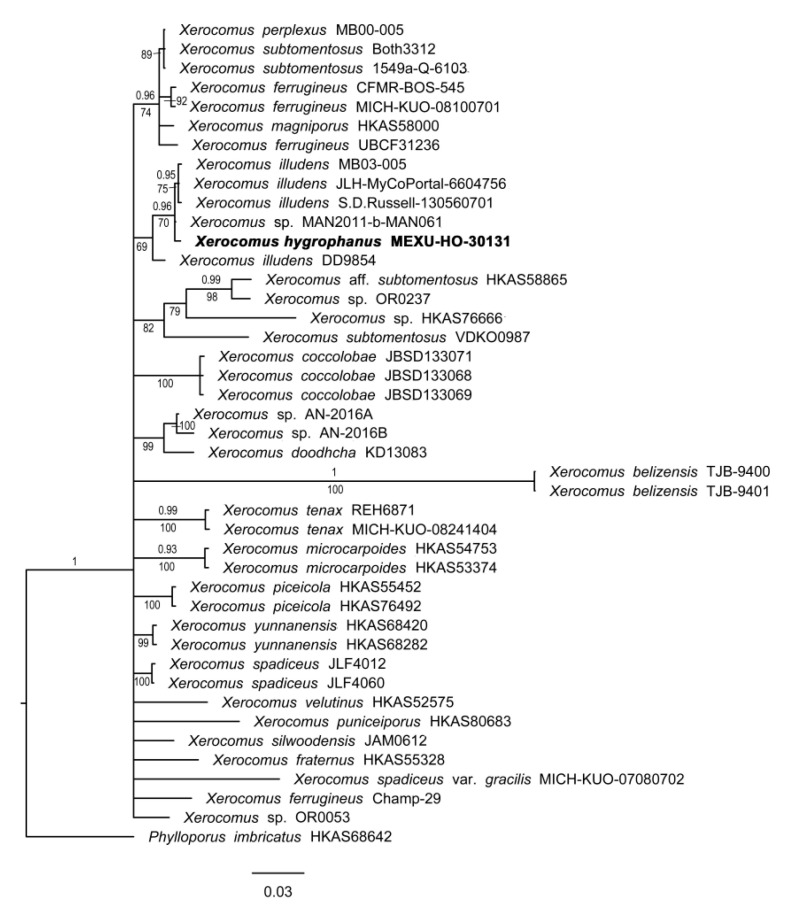
Phylogram of *Xerocomus* with ITS, nrLSU, and RPB2. Maximum likelihood (ML)/Bayesian inference (BI) analyses were performed. The phylogram presents Bayesian inference, bootstrap frequencies (≥50%), and posterior values (BI ≥ 0.90) at the supported branches. *Phylloporus imbricatus* was used as the outgroup. Newly generated sequences are indicated in bold.

**Figure 7 jof-09-01126-f007:**
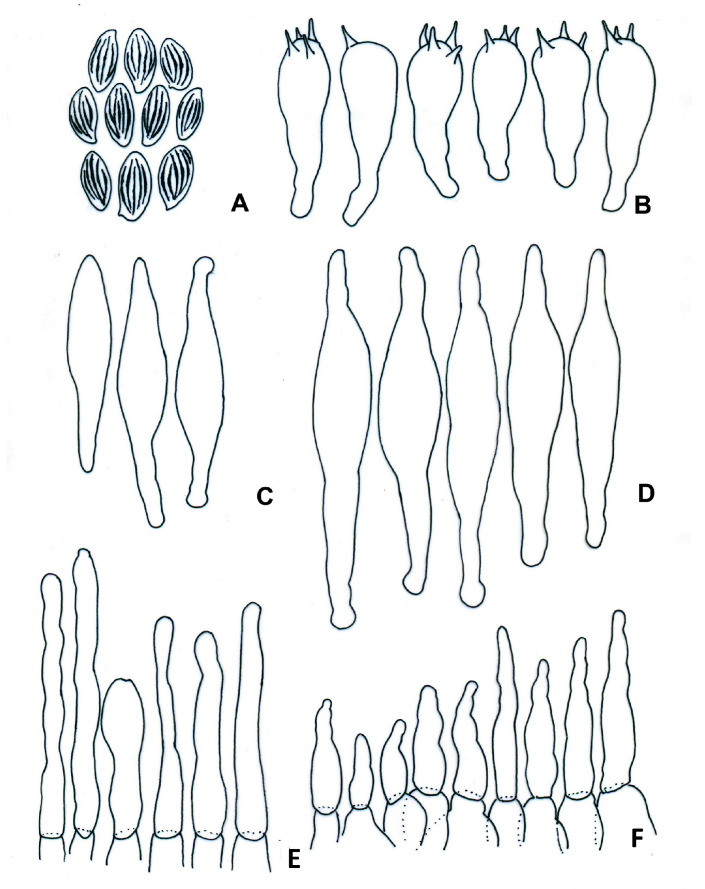
*Boletellus minimatenebris* (1076-ITCV, MEXU-HO-30119, holotype): (**A**) basidiospores; (**B**) basidia; (**C**) cheilocystidia; (**D**) pleurocystidia; (**E**) terminal elements of the pileipellis; and (**F**) caulocystidia. Scale bars: 10 µm.

**Figure 8 jof-09-01126-f008:**
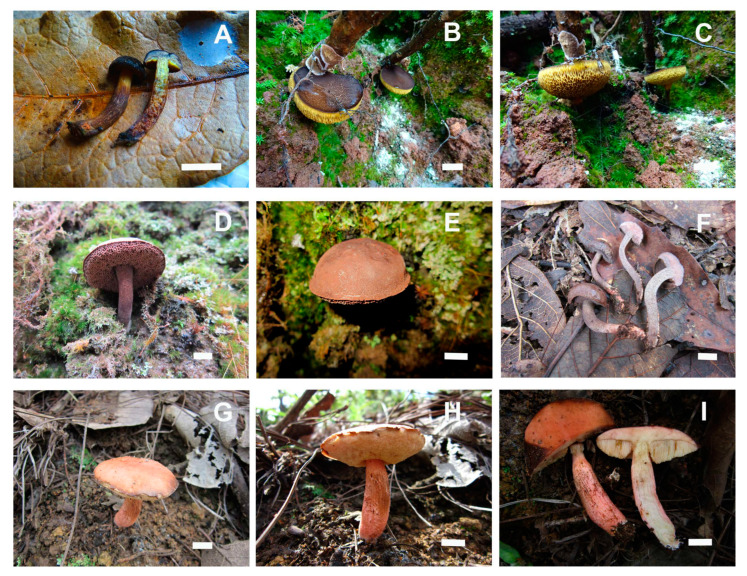
(**A**–**C**) *Boletellus minimatenebris* (1076-ITCV, MEXU-HO-30119); (**D**–**F**) *Cacaoporus mexicanus* (ITCV-904, MEXU-HO-30110); and (**G**–**I**) *Garcileccinum salmonicolor* (MEXU-HO:30108). Scale bars: 10 mm.

**Figure 9 jof-09-01126-f009:**
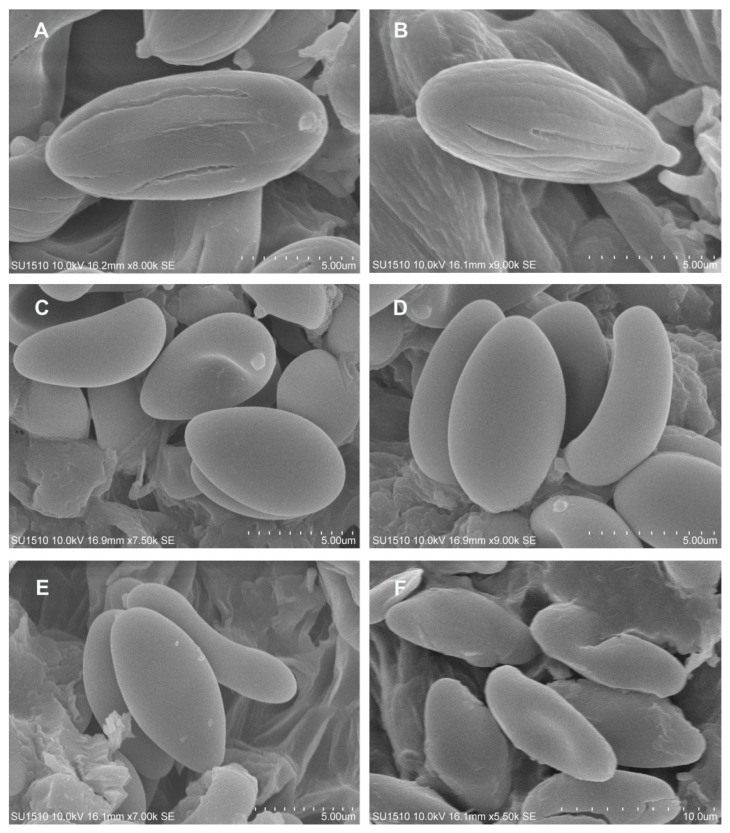
Basidiospores under the scanning electron microscope (SEM): (**A**,**B**) *Boletellus minimatenebris*; (**C**,**D**) *Cacaoporus mexicanus*; and (**E**,**F**) *Xerocomus hygrophanus*.

**Figure 10 jof-09-01126-f010:**
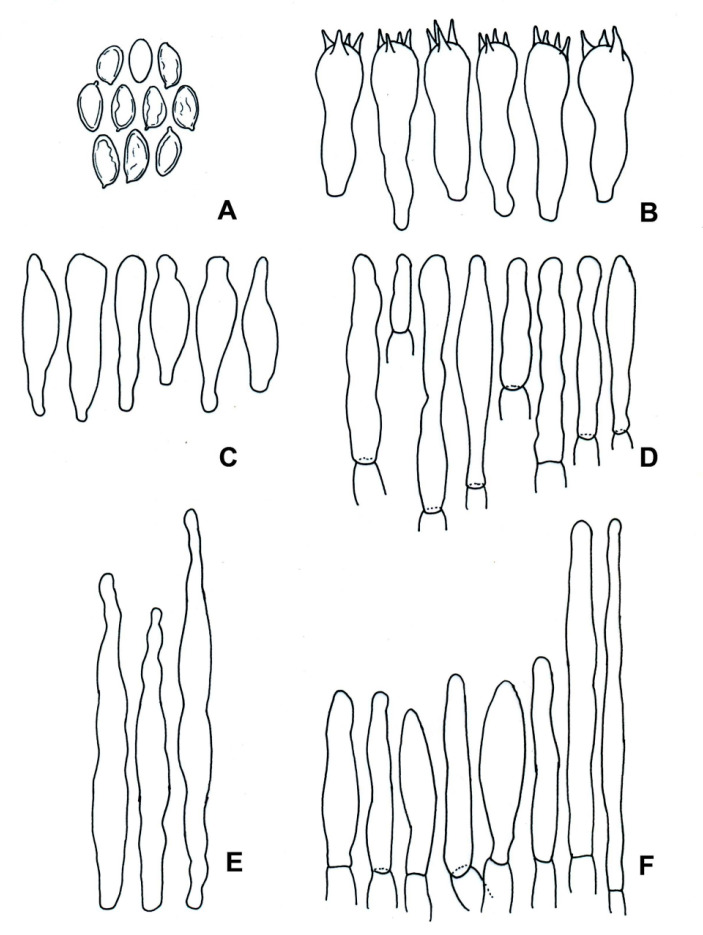
*Cacaoporus mexicanus* (ITCV-870, MEXU-HO-30122), holotype: (**A**) basidiospores; (**B**) basidia; (**C**) cheilocystidia; (**D**) pleurocystidia; (**E**) terminal elements of the pileipellis; and (**F**) caulobasidia. Scale bars: 10 µm.

**Figure 11 jof-09-01126-f011:**
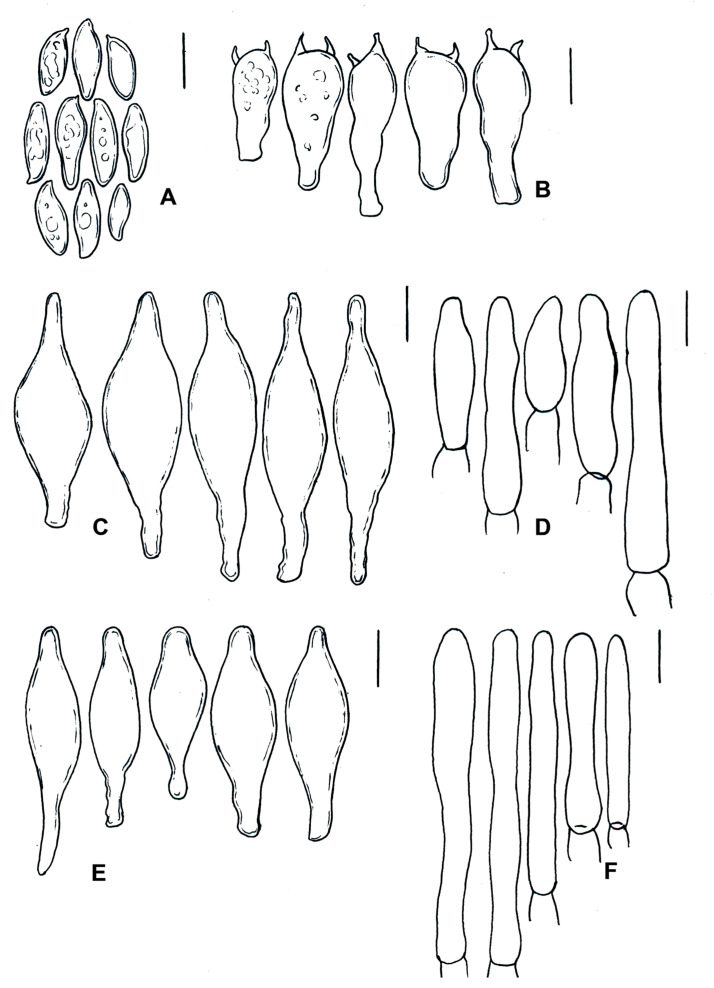
*Garcileccinum salmonicolor* (MEXU-HO-30108): (**A**) basidiospores; (**B**) basidia; (**C**) cheilocystidia; (**D**) pleurocystidia; (**E**) caulobasidia; and (**F**) terminal elements of the pileipellis. Scale bars: 10 µm.

**Figure 12 jof-09-01126-f012:**
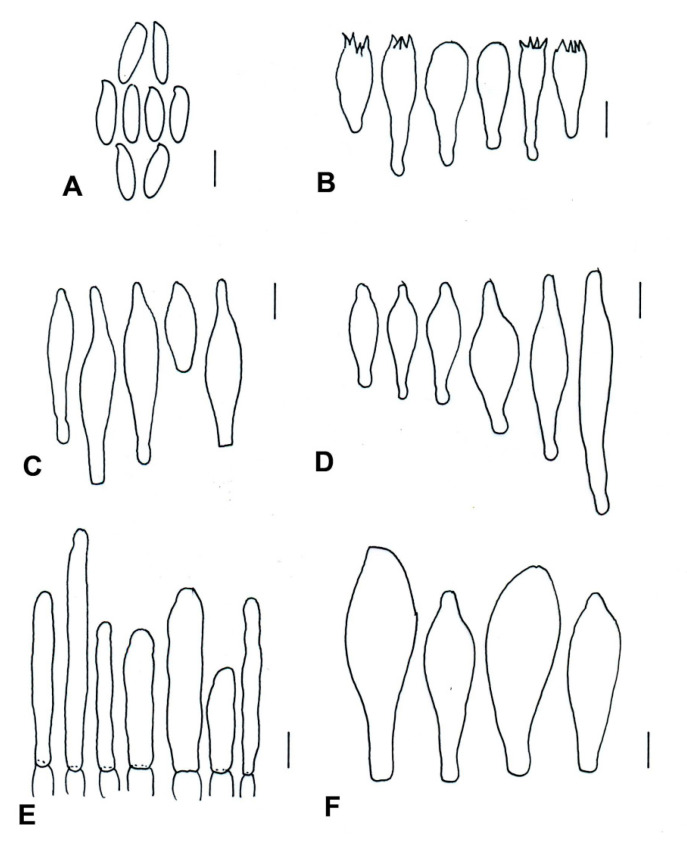
*Leccinum oaxacanum* (MEXU HO 30460, holotype): (**A**) basidiospores; (**B**) basidia; (**C**) cheilocystidia; (**D**) pleurocystidia; (**E**) terminal elements of the pileipellis; and (**F**) caulocystidia. Scale bars: 10 µm.

**Figure 13 jof-09-01126-f013:**
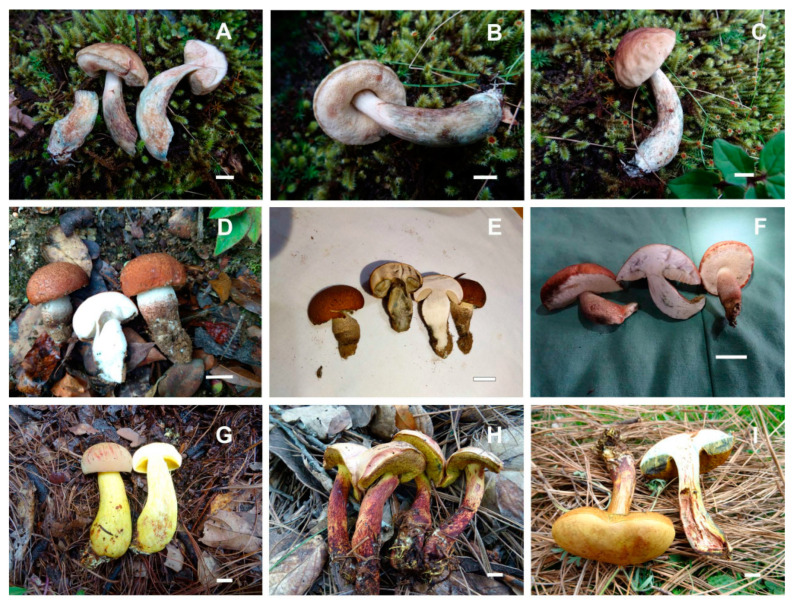
(**A**–**C**) *Leccinum juarenzense* (holotype, MEXU-HO:30112); (**D**–**F**) *Leccinum oaxacanum* (holotype, MEXU-HO-30460); and (**G**–**I**); *Pulchroboletus neoregius* (MEXU-HO-30424, MEXU-HO-30425). Scale bars = 10 mm.

**Figure 14 jof-09-01126-f014:**
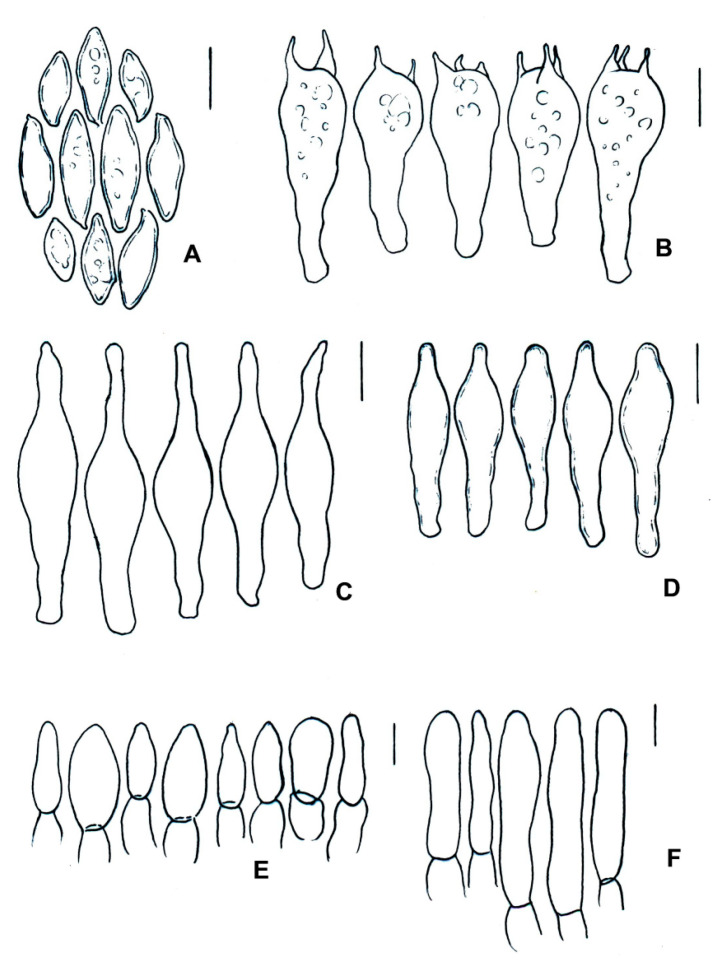
*Leccinum juarenzense* (MEXU HO, holotype): (**A**) basidiospores; (**B**) basidia; (**C**) cheilocystidia; (**D**) pleurocystidia; (**E**) terminal elements of the pileipellis; and (**F**) caulocystidia. Scale bars: 10 µm.

**Figure 15 jof-09-01126-f015:**
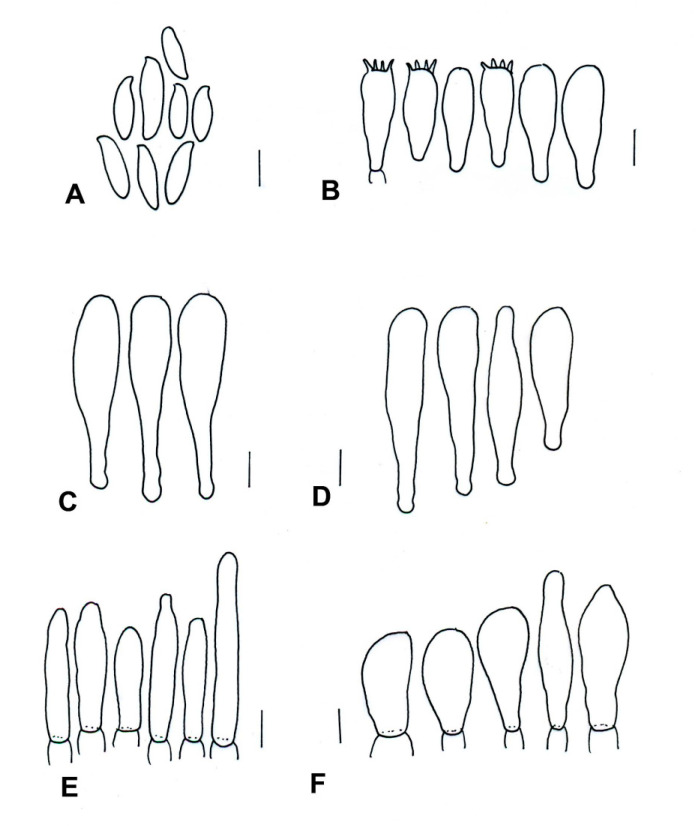
*Pulchroboletus neoregius* (MEXU-HO 30425)*:* (**A**) basidiospores; (**B**) basidia; (**C**) pleurocystidia; (**D**) cheilocystidia; (**E**) terminal elements of the pileipellis; and (**F**) caulocystidia. Scale bars: 10 µm.

**Figure 16 jof-09-01126-f016:**
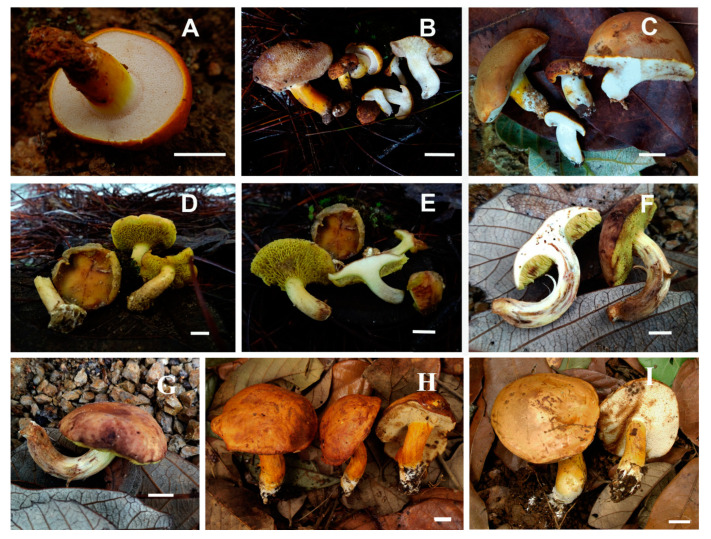
(**A**–**C**) *Tylopilus pseudoleucomycelinus* (holotype, MEXU-HO-30115); (**D**–**G**) *Xerocomus hygrophanus* (holotype, MEXU-HO-30131); and (**H**,**I**) *Tylopilus leucomycelinus* (M. Caballero 53, César 64). Scale bars A–L: 10 mm.

**Figure 17 jof-09-01126-f017:**
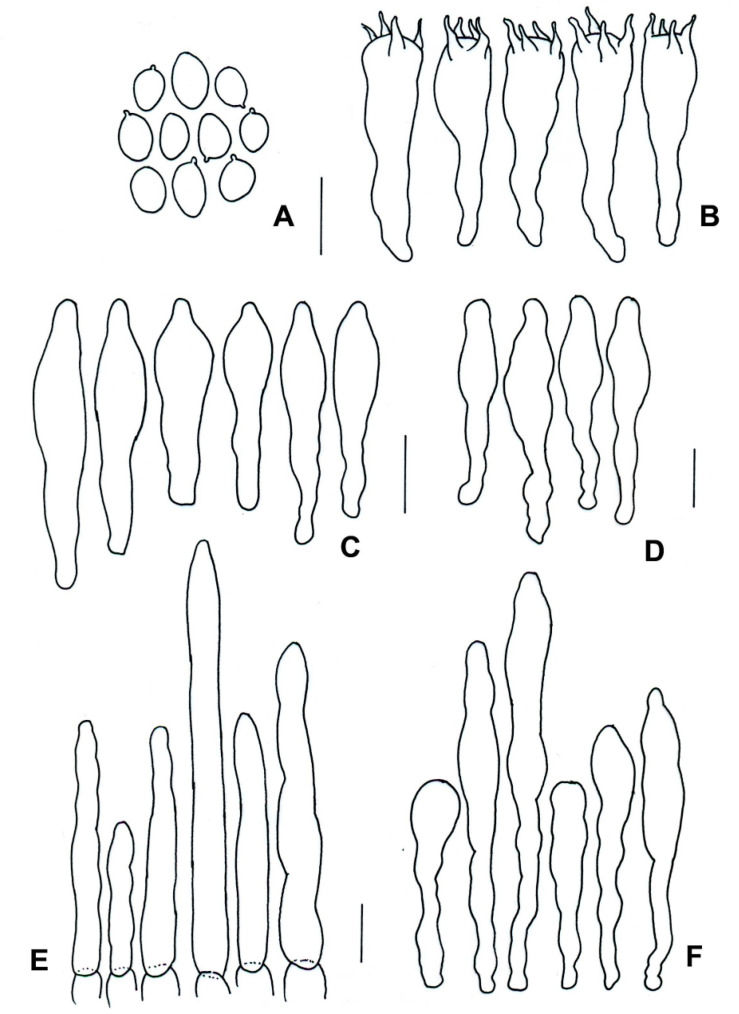
*Tylopilus pseudoleucomycelinus* (ITCV 1074, MEXU-HO-30115 duplicate): (**A**) basidiospores; (**B**) basidia; (**C**) cheilocystidia; (**D**) pleurocystidia; (**E**) terminal elements of the pileipellis; and (**F**) caulocystidia. Scale bars: 10 µm.

**Figure 18 jof-09-01126-f018:**
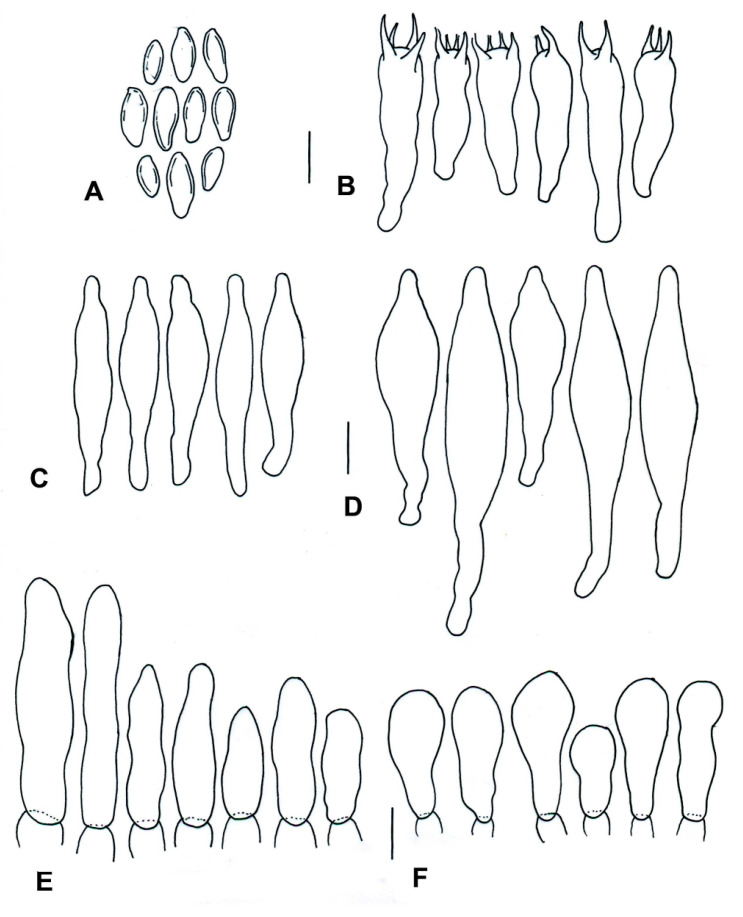
*Xerocomus hygrophanus* (MEXU-HO-30131, holotype): (**A**) basidiospores; (**B**) basidia; (**C**) pleurocystidia; (**D**) cheilocystidia; (**E**) pileipellis; and (**F**) caulobasidia. Scale bars: 10 µm.

**Table 1 jof-09-01126-t001:** Specimens and sequences used for the molecular phylogenetic analyses in Figure 1. Sequences newly generated for this study are highlighted in bold.

Fungal Taxon	Specimen Voucher	ITS	nrLSU	RPB2
*Boletellus ananas*	REH8613	------------------------	KP327629	-------------------
*B. ananas*	CFMR BZ-3167	------------------------	MK601717	MK766279
*B. ananas*	NY815459	------------------------	JQ924336	KF112760
*B. ananas*	NY796159	------------------------	JQ924335	-------------------
*B. ananiceps*	REH9484	------------------------	KP327655	-------------------
*B. ananiceps*	3794	------------------------	KP327666	-------------------
*B. aurocontextus*	N.K.Zeng2536	MT822935	MT829120	-------------------
*B. belizensis*	CFMR:BZ-316 TJB-9128	MN250194	MN250169	-------------------
*B. belizensis*	CFMR BZ-316 TJB-9128	NG_078671	------------------------	-------------------
*B. belizensis*	CFMR:BZ-429 JCB-2001-238	MN250210	MN250185	-------------------
*B. belizensis*	CFMR:BZ-1564 BOS-218	MN250209	MN250184	-------------------
*B. badiovinosus*	REH8923	KP327640	------------------------	-------------------
*B. brunoflavus*	B21061253	ON364057	ON195936	ON390839
*B. brunoflavus*	GDGM87995	ON364056	ON195935	-------------------
*B. chrysenteron*	54/97	DQ534634	------------------------	-------------------
*B. chrysenteron*	REH9015	------------------------	KP327645	-------------------
*B. chrysenteron*	iNaturalist # 91541460	OM809388	OM809388	-------------------
*B. chrysenteron*	HKAS105270	MT520080	MT520080	-------------------
*B. indistinctus*	HKAS 77623	------------------------	NG_058689	KT990371
*B. indistinctus*	HKAS80681	------------------------	KT990532	KT990368
*B. obscurococcineus*	HKAS77662	------------------------	------------------------	KT990372
*B. obscurococcineus*	isolate: 3A3	AB973769	AB973769	-------------------
** *B. minimatenebris* **	**MEXU-HO-30119; ITCV1076**	**OR713119**	**OR713121**	**OQ938895**
** *B. minimatenebris* **	**ITCV-1087**	**OR713120**	**OR713122**	**OQ938894**
*B. pseudochrysenteroides*	Mushroom Observer #288188	------------------------	MH257564	-------------------
*B. sinapipes*	REH9018	------------------------	KP327647	-------------------
*B. sinapipes*	REH9408	------------------------	KP327653	-------------------
*B. shoreae*	AP6679	MH608209	MH608211	-------------------
*B. shoreae*	AP6696	MH608208	MH608210	-------------------
*B. wenshanensis*	HKAS 122938	------------------------	NG_154028	-------------------
*B. wenshanensis*	532624MF_103_Wu2281	------------------------	ON006512	ON007376
*Boletellus* sp.	HKAS 74888	KF112413	------------------------	KF112747
*Boletellus* sp.	HKAS 80554	------------------------	KT990535	KT990535
*Boletellus* sp.	HKAS122526	ON794351	------------------------	-------------------
*Boletellus* sp.	TAA195080	AM412293	------------------------	-------------------

**Table 2 jof-09-01126-t002:** Specimens and sequences used for the molecular phylogenetic analyses in Figure 2. Sequences newly generated for this study are highlighted in bold.

Fungal Taxon	Specimen Voucher	RPB2	ATP6
** *Cacaoporus mexicanus* **	**ITCV-1023, MEXU-HO 30103**	**OQ938899**	------------------------
** *C. mexicanus* **	**ITCV-904, MEXU-HO-30110**	**OQ938897**	------------------------
** *C. mexicanus* **	**ITCV-978, MEXU-HO 30128**	**OQ938898**	**OR683446**
** *C. mexicanus* **	**ITCV-1027**	**OQ938896**	**OR683445**
*C. pallidicarneus*	OR0681	MK372283	MK372259
*C. pallidicarneus*	OR0683	MK372284	MK372260
*C. pallidicarneus*	OR1306	MK372285	MK372261
*C. pallidicarneus*	SV0221	MK372286	MK372262
*C. pallidicarneus*	SV0451	MK372287	MK372263
*C. tenebrosus*	OR0654	MK372288	MK372265
*C. tenebrosus*	OR1435	MK372289	MK372265
*C. tenebrosus*	SV0223	MK372290	MK372266
*C. tenebrosus*	SV0224	MK372291	MK372267
*C. tenebrosus*	SV0452	MK372292	MK372269
*Cacaoporus* sp.	SV0402	MK372293	MK372270
*Cupreoboletus poikilochromus*	GS10070	KT157068	-----------
*Cu. poikilochromus*	GS11008	KT157067	-----------
*Cyanoboletus bessettei*	ARB1393B	MW737458	-----------
*Cy. bessettei*		MW737457	-----------
*Cy. brunneoruber*	HKAS80579_1	KT990401	-----------
*Cy. brunneoruber*	OR0233	MG212628	MG212542
*Cy. cyaneitinctus*	Farid_340	MW737461	-----------
*Cy. cyaneitinctus*	Farid_920	MW737465	-----------
*Cy. cyaneitinctus*	JAB_324	MW737469	-----------
*Cy. cyaneitinctus*	JAB_184	MW737467	-----------
*Cy. instabilis*	HKAS59554	KF112698	-----------
*Cy. pulverulentus*	RW109	KT824013	KT823980
*Cy. sinopulverulentus*	HKAS59609	KF112700	-----------
*Cyanoboletus* sp.	OR0257	MG212629	MG212543
*Cyanoboletus* sp.	OR0322	MH614768	MH614673
*Cyanoboletus* sp.	OR0491	MT823996	MH614674
*Cyanoboletus* sp.	OR0961	MH614770	MH614675
*Chalciporus africanus*	JD517	KT823996	KT823963
*Lanmaoa angustispora*	HM605178	KM605178	-----------
*La. angustispora*	HKAS74765	KF112680	-----------
*La. asiatica*	HKAS63516	KT990419	-----------
*La. asiatica*	OR0228	MH614777	MH614682
*La. carminipes*	BOTH4591	MG897439	MG897419
*La. flavorubra*	NY775777	KF112681	-----------
*La. pallidorosea*	BOTH4432	MG897437	MG897417
*Lanmaoa* sp.	OR0130	MH614778	MH614683
*Lanmaoa* sp.	OR0370	MH614779	MH614684
*Rubinoboletus rubinus*	AF2835	KT823995	KT823962

**Table 3 jof-09-01126-t003:** Specimens and sequences of Leccinoideae subfamily used for the molecular phylogenetic analyses in Figure 3. Sequences newly generated for this study are highlighted in bold.

Fungal Taxon	Specimen Voucher	nrLSU	RPB2	TEF1
** *Garcileccinum salmonicolor* **	**MEXU-HO-30108**	**OQ909093**	**OQ938917**	**OR683442**
** *G. salmonicolor* **	**MEXU-HO-30105**	**OQ909094**	**OQ938918**	**OR683443**
** *G. violaceotinctum* **	**CFMR-BZ-1676**	--------------------	**QGW57932**	**QGW57818**
*G. violaceotinctum*	CFMR-BZ-3169	--------------------	QGW57933	QGW57819
*G. viscosum*	NY 01194025	--------------------	QGW57910	QGW57795
*G. viscosum*	CFMR-BZ-2406	-------------------	-----------------	-----------------
*Hemileccinum rugosum*	HKAS 84355	KT990578	-----------------	-----------------
*Lanmaoa angustispora*	HM605178	--------------------	KM605178	-----------------
*La. angustispora*	HKAS74765	--------------------	KF112680	-----------------
*La. asiatica*	HKAS63516	--------------------	KT990419	-----------------
*La. asiatica*	OR0228	--------------------	MH614777	-----------------
*La. carminipes*	BOTH4591	--------------------	MG897439	-----------------
*La. flavorubra*	NY775777	--------------------	KF112681	-----------------
*La. pallidorosea*	BOTH4432	--------------------	MG897437	-----------------
*Lanmaoa* sp.	OR0130	--------------------	MH614778	-----------------
*Lanmaoa* sp.	OR0370	--------------------	MH614779	-----------------
*Leccinellum albellum*	MICH-KUO-09200807	MK601747	MK766309	MK721101
*Le. albellum*	MICH-KUO-07241101	MK601746	MK766308	MK721100
*Le. alborufescens*	FHMU 1758	--------------------	MK816332	MK816330
*Le*. *alborufescens*	FHMU1908	--------------------	MK816333	MK816330
*Le*. *citrinum*	HKAS53427	KF112488	KF112727	KF112253
*Le. citrinum*	HKAS53410	KT990585	KT990421	-----------------
*Le. crocipodium*	MICH-KUO-07050707	MK601749	MK766311	MK721103
*Le. pseudoscabrum*	MICH-KUO-10200710	MK601753	MK766314	MK721107
*Le. pseudoscabrum*	CFMR-DPL-11432	MK601752	MK766313	MK721106
*Le. quercophilum*	NY01293386	--------------------	-----------------	-----------------
*Leccinum* aff. *scabrum*	HKAS57266	KF112442	KF112722	KF112248
*L. album*	Li1072	MZ392872	MW439260	MW439267
*L. aurantiacum*	L0342207	MK601759	MK766318	MK721113
*L. duriusculum*	Yang5971	MZ675541	MZ707779	MZ707785
*L. holopus*	MICH-KUO-09150707	MK601763	MK766322	MK721117
*L. holopus*	CFMR-BOS-569	MK601762	MK766321	MK721116
*L. holopus*	Yang 5972	--------------------	MW439258	--------------------
*L. juarenzense*	MEXU HO 30,112	OQ909095	OQ938916	--------------------
*L. manzanitae*	NY14041	MK601765	MK766324	MK721119
*L. melaneum*	FB491	MZ675542	MZ707780	MZ707786
*L. monticola*	NY00815448	HQ161869	HQ161838	-----------------
*L. monticola*	NY760388	MK601766	-----------------	-----------------
*L. monticola*	HKAS76669	KF112443	KF112723	KF112249
*L. oaxacanum*	MEXU-HO-30460	OQ909096	OQ938915	OR683444
*L. parascabrum*	Li1700	MW413912	MW439265	MW439272
*L. parascabrum*	Wu1784	--------------------	MW439264	MW439271
*L. pseudoborneense*	WGS960	MW413909	MW439262	MW439270
*L. pseudoborneense*	WGS965	MW413910	MW439263	MW439269
*L. pseudoborneense*	Zhao773	--------------------	MZ543309	MZ543307
*L. pseudoborneense*	Zhao476	--------------------	MZ543308	MZ543306
*L. pseudoborneense*	WGS947	MW413908	MW439261	MW439268
*L. quercinum*	HKAS63502	KF112444	KF112724	KF112250
*L. scabrum*	HKAS53604	KT990586	KT990422	---------------------
*L. scabrum*	HKAS56371	KT990587	KT990423	KT990782
*L. snellii*	CFMR-BOS-579	MK601773	MK766331	MK721127
*L. tablense*	NY00796175	MK601775	MK766333	MK721129
*L. tablense*	NY760387	MK601774	MK766332	MK721128
*L. variicolor*	HKAS57758	KF112445	KF112725	KF112251
*L. versipelle*	CFMR-DLC2002-122	MK601778	MK766336	MK721132
*L. versipelle*	FB27	MZ675546	MZ707782	MZ707790
*L. versipelle*	LJW418	MZ675545	MZ707781	MZ707789

**Table 4 jof-09-01126-t004:** Specimens and sequences used for the molecular phylogenetic analyses in Figure 4. Sequences newly generated for this study are highlighted in bold.

Fungal Taxon	Specimen Voucher	ITS	nrLSU
*Alessioporus ichnusanus*	MG 549a	KJ729493	KJ729506
*A. rubriflavus*	JLF2561	KC812305	KC812306
** *Pulchroboletus neoregius* **	**MEXU-HO-30423**	-------------------------------	**OQ940034**
** *P. neoregius* **	**MEXU-HO-30424**	**OQ940040**	------------------------
** *P. neoregius* **	**MEXU-HO-30425**	**OQ940039**	**OQ940035**
** *P. neoregius* **	**MEXU-HO-30426**	**OQ940040**	**OQ940033**
*P. neoregius*	NY796166	JQ924304	JQ924344
*P. neoregius*	NY575822	JQ924303	JQ924343
*P. roseoalbidus*	AMB 12757	NR_154305	NG_060126
*P. roseoalbidus*	MCVE 17577	KJ729490	KJ729503
*P. roseoalbidus*	MCVE 18217	KJ729488	KJ729501
*P. roseoalbidus*	MG416a	KJ729489	KJ729502
*P. roseoalbidus*	MG532a	KJ729487	KJ729500
*P. rubricitrinus*	Farid335	MF193884	MG026638
*P. sclerotiorum*	Flas-F-60333	MF098659	MF614166
*P. sclerotiorum*	Mushroom Observer #243879	MH257550	MH257545
*Pulchroboletus* sp.	JLF5434	MH213050	MH201325
*Pulchroboletus* sp.	JLF6328	MH213051	MH201326
*Pulchroboletus* sp.	JLF6329	MH213053	MH201328

**Table 5 jof-09-01126-t005:** Specimens and sequences used for the molecular phylogenetic analyses in Figure 5. Sequences newly generated for this study are highlighted in bold.

Fungal Taxon	Specimen Voucher	ITS	nrLSU	RPB2
*Tylopilus* aff. *balloui*	1D10	AB973734	-------------	-------------
*T.* aff. *balloui*	Mushroom 247674	KY859806	-------------	-------------
*T. argillaceus*	HKAS90186	-------------	KT990589	KT990424
*T. atroviolaceobrunneus*	HKAS84351	-------------	KT990625	MT110421
*T. aurantiacus*	Osmundson 1198	-------------	EU430740	---------------
*T. aurantiacus*	HKAS59700	-------------	NG081276	KF112740
*T. aurantiacus*	NY2072258	-------------	EU430740	-------------
*T. balloui*	CMU51-SL-42	KX017307	KX017298	-------------
*T. balloui*	CMU51-SL-39	KX017306	KX017297	-------------
*T. balloui*	CMU51-SL-37	KX017305	KX017296	-------------
*T. balloui*	CMU51-SL-32	KX017304	KX017295	-------------
*T. balloui*	2D7	AB973758	-------------	----------------------
*T. balloui*	2D6	AB973757	-------------	----------------------
*T. balloui*	1D9	AB973733	-------------	----------------------
*T. balloui*	1E1	AB973735	-------------	----------------------
*T. balloui*	FMNH 1073250	-------------	EU430733	----------------------
*T. balloui*	Osmundson 1121	-------------	EU430743	EU434333
*T. balloui*	Osmundson 1117	-------------	EU430741	----------------------
*T. balloui*	Osmundson 1105	-------------	-------------	----------------------
*T. balloui*	HKAS51151	-------------	KT990590	KT990425
*T. formosus*	PDD72637	HM060320	HM060319	----------------------
*T. griseipurpureus*	MG521a	KM975484	KM975493	----------------------
*T. himalayanus*	DC 17-25	MG799322	MG799328	----------------------
*T. leucomycelinus*	DR2800	MN115814	MN115804	----------------------
*T. leucomycelinus*	JBSD127420	MN115813	MN115803	MN095210
*T. leucomycelinus*	JBSD127419	MN115812	MN115802	MN095209
** *T. leucomycelinus* **	**ECC 64**	**OQ984890**	-------------	----------------------
** *T. leucomycelinus* **	**GH111**	**OQ984889**	-------------	----------------------
*T. leucomycelinus*	BZ2409	MN115815	MN115805	----------------------
** *T. leucomycelinus* **	**MC53**	**OQ984888**	-------------	----------------------
*T. leucomycelinus*	18463	JF908789	-------------	----------------------
*T. leucomycelinus*	Halling 8526	-------------	EU430736	EU434336
*T. leucomycelinus*	Halling 8521	-------------	EU430735	----------------------
** *T. leucomycelinus* **	**JC1298**	**OQ984887**	-------------	----------------------
*T. leucomycelinus*	F1030852	MN115811	-------------	----------------------
*T. microspores*	HMAS84730	NR137924	NG059538	----------------------
*T. neofelleus*	DC 16-64	MG777524	MG777529	----------------------
*T. neofelleus*	YT20120811	KM975487	KM975495	----------------------
*T. otsuensis*	HKAS53401		KF112449	KF112797
*T. plumbeoviolaceoides*	GDGM:42624	-------------	KM975498	----------------------
*T. plumbeoviolaceoides*	Chu25	DQ407261	-------------	----------------------
*T. plumbeoviolaceoides*	GDGM43255	KM975490	-------------	----------------------
*T. plumbeoviolaceoides*	GDGM20311	NR137601	NG059490	----------------------
*T. porphyrosporus*	17898	JF908788		----------------------
*T. pseudoballoui*	DC 17-30	MG799329	MG799327	----------------------
*T. pseudoballoui*	DC 17-35	MG799324	MG799325	----------------------
** *T. pseudoleucomycelinus* **	**MEXU-HO-30115**	**OQ940043**	**OQ940037**	**OQ938903**
*T. pseudoscaber*	DAVFP27804	JF899578	-------------	----------------------
*T. rubrobrunneus*	F-PRL7359	GQ166869	-------------	----------------------
*T. rubrotinctus*	KK259	---------------	MT154733	MW165283
*T. rubrotinctus*	LF1101	--------------	MT154732	MW165291
*T. sordidus*	JMP0094	EU819450	--------------------	----------------------
*T. virens*	HE2757	KC505585	--------------------	----------------------

**Table 6 jof-09-01126-t006:** Specimens and sequences used for the molecular phylogenetic analyses in Figure 6. Sequences newly generated for this study are highlighted in bold.

Species	Voucher	ITS	LSU	RPB2
*Xerocomus coccolobae*	JBSD133071 (ANGE1405)	OQ108302	OQ102365	OQ117434
*X. coccolobae*	JBSD133068 (ANGE915)	OQ108300	OQ102363	OQ117435
*X. coccolobae*	JBSD133069 (ANGE965)	OQ108301	OQ102364	OQ117436
** *X. hygrophorus* **	**MEXU-HO-30131**	**OQ984883**	**OQ975751**	**OQ938901**
*X. illudens*	JLH MyCoPortal 6604756	MK578706	MK578706	-------------
*X. illudens*	DD9854	-------------	AY612840	-------------
*X. illudens*	MB03-005	JQ003658	JQ003658	-------------
*X. illudens*	iNaturalist # 130560701	OP643377	OP643377	-------------
*X. ferrugineus*	CFMR BOS 545	-------------	MK601819	MK766375
*X. ferrugineus*	KUO-08100701	-------------	MK601820	MK766376
*X. ferrugineus*	UBC:F31236	MZ817040	MZ817040	-------------
*X. ferrugineus*	Champ-29	KX449436	KX449436	-------------
*X. fulvipes*	HKAS 76666	--------------	KF112390	KF112789
*X. magniporus*	HKAS 58000	--------------	KF112392	KF112781
*X. puniceiporus*	HKAS 80683	--------------	KU974141	KU974146
*X. perplexus*	MB00-005	--------------	JQ003702	-------------
*X. rugosellus*	HKAS 58865	--------------	KF112389	KF112784
*Xerocomus* sp.	MAN061	JQ003656	JQ003707	-------------
*Xerocomus* sp.	MAN061	JQ003656	------------	-------------
*Xerocomus* sp.	OR0053	-------------	------------	MH580834
*Xerocomus* sp.	OR0237			MH580835
*X. spadiceus*	MICH KUO 7080702	-------------	MK601822	MK766378
*X. spadiceus*	JLF4012	KX534078	--------------	--------------
*X. spadiceus*	JLF4060	KX534079	--------------	--------------
*X. silwoodensis*	JAM0612	-------------	KF030323	--------------
*X. subtomentosus*	VDKO0987	-------------	--------------	MG212657
*X. subtomentosus*	Xs1	AF139716	--------------	--------------
*X. subtomentosus*	Both3312	DQ066413	--------------	--------------
*X. subtomentosus*	1549a-Q-6103	KM248935	--------------	--------------
*X. tenax*	REH6871	-------------	KF030320.1	--------------
*X. tenax*	MICH KUO 8241404	-------------	MK601823	MK766379
*X. tenax*	REH6871	--------------	KF030320	--------------
*Phylloporus imbricatus*	HKAS84355	NG_059621	KT990578	KT990413

**Table 7 jof-09-01126-t007:** Comparative morphology of species in *Tylopilus* complex *balloui*.

Species	Basidiomata	Basidiospores	Cystidia or Pseudocystidia	Terminal Elements of the Pileipellis	Caulobasidia
*T. balloui* *	40–85 mm.	8.2–10.5 × 3–3.7 μm, ellipsoid, ovoid to fusoid.	27–60 (−80) × 6– 21 μm, utriform.	4–8 μm, elongated cylindrical.	Clavate dermatocystidia, 33 × 14 μm.
*T. leucomycelinus* **	(36–) 43–72 (–110) mm.	6.6 (–8.2) × (3.9–) (–5.5) μm, allantoid, phaseoliform, ovoid, subglobose.	(38–) 42–75 (–83) × 7–16 (–19) μm, fusiform to ventricose fusiform or lageniform to sublageniform.	24–115 × 3–8 μm, cylindrical.	40–75 (–88) × 7–18 (–20) μm, fusiform to ventricose fusiform or lageniform to sublageniform.
*T. pseudoleucomycelinus* ***	19–28 (40) mm.	5–7 (–10) × 4–5 µm ovoid to lachrymoid, ellipsoid.	30–35 (−66) × 7–8 (−12) µm, fusiform, subclavate to piriform.	30–40 × 7–9 µm, cylindrical.	Caulocystidia 25–60 × 7–11 µm, fusiform, ventricose-rostrate.

Based on * Singer [81]; ** Gelardi et al. [26]; and *** this work.

## Data Availability

Data are contained within the article.

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
