# Peer review of "Broadening the Knowledge of Mexican Boletes: Addition of a New Genus, Seven New Species, and Three New Combinations"

_jof, 2023, doi:10.3390/jof9121126_

Round 1

Reviewer 1 Report

Comments and Suggestions for Authors

In this manuscript, one new genus, six new species and three new combinations are proposed on Boletoid mushrooms. Lacunae noticed by me are given below.

1) At least two to three protein coding markers are required to erect a new genus. I suggest authors to rerun their phylogeny with at least LSU-RPB2-TEF1 and then come to the conclusion for erecting Garcileccinum.

2) Without considering ITS region a species of Xerocomus can not be revealed as most of the species in Xerocomus are established with ITS. Combination of LSU-RPB2 is not sufficient for the same.

3) What do you mean by writing Melzer solution's? Correct every where with Melzer's reagent.

4) Drawing are poor, very inconsistent and not suitable to be published in JOF. I would suggest to refer an article like below on Boletaceae that is also published in the same journal (JOF) very recently:

DAS, K.; GHOSH, A.; CHAKRABORTY, D.; DATTA, S.; BERA, I.; LAYOLA MR, R.; BANU, F.; VIZZINI, A.; WISITRASSAMEEWONG, K. 2023. Four Novel Species and Two New Records of Boletes from India. Journal of Fungi 9, 754. https://doi.org/10.3390/jof9070754

5) Use your aesthetic sense and rotate the sub-plates (F & H) of Figure  15.

6) Voucher table including the sequence of the used markers is needed. 

7) Some of my suggestions are incorporated directly in PDF

Comments on the Quality of English Language

I have hardly seen such a poor English language in any scientific article. Generally in scientific description we do not use be verb (is/am/are) while in writing notes we use proper English. Always add adjective before Noun. Authors did not have any sense of writing English. Every NOTES must be edited thoroughly and very seriously by a native English speaker otherwise this manuscript should not be considered for a publication. Presently, English is just horrible throughout the manuscript.

Author Response

JOURNAL OF FUNGI

Manuscript ID:  jof- 2636339 
Type of manuscript: Article

Title: Broadening the knowledge of Mexican Boletes: Addition of a new genus, seven new species and three new combinations

We deeply appreciate the valuable corrections, suggestions and comments from the editor and the three reviewers. In the updated version, now we have taken into account all the suggestions which in our humble opinion substantially enhance the manuscript quality, including a careful check of the English. Taking into account the valuable suggestions provided we have included now our specific answers in the following table. Additionally, we have highlighted with yellow text in the updated manuscript the additions or changes we have made to answer the editor´s and reviewers’ requests. Thanks a lot for your professionalism dealing with our manuscript.

The lines mentioned correspond to the PDF version received from reviewers 1 and 2, in the attached file.

Reviewer 2 Report

Comments and Suggestions for Authors

The reviewer have commented in detail with the pdf file uploaded. The manuscript contains a new genus, six new species and three new combinations. The pictures are high in resolution. The descriptions are detailed and accurated. However, there are still some work to do before this manuscript can be accepted. The reviewer have selected some important comment from the pdf file as following.

Introduction:

It is better provide the morphological concepts of Boletaceae and related families.

The infrastructure taxonomy of Bolataceae can be introduced here. Which subfamilies are included in Bolataceae.  

The conept of Leccinoideae subfamily is also needed.

Representative tree species of Mexico Boletes growing areas should be provided.

Materials and Methods

Referential sequences GenBank numbers should be provided. Also references for these sequences. I did not find these informations.

Abbreviations of author names of the Bolataceae members (genera and species) can be added when they appeared in the manuscript for the first time. The authors should check this point for all parts of this manuscript.

Results

It is better label these clades as A, B, C... in corresponding figures.

Type specimens of this analysis should be labeled in figures.

Specimens of different continents can be presented in different colors. Thus the geological impact for phylogenetic topology can be shown.

Type species of the new genus should be labeled.

The line drawings of this only shows several terminal elements of pileipellis (or suprapellis). It must contain the orientation of the mycelia.

Not very clearly for the thick wall elements.

Styles of line drawings are not the same in this manuscript.

Discussion

Current form of the discussion only contains references of the previous articles/books. The readers may expect some contents contributed by the authors.

Comments on the Quality of English Language

The manuscirpt is generally well written. Some minor improvement are needed.

Author Response

JOURNAL OF FUNGI

Manuscript ID:  jof- 2636339 
Type of manuscript: Article

Title: Broadening the knowledge of Mexican Boletes: Addition of a new genus, seven new species and three new combinations

We deeply appreciate the valuable corrections, suggestions and comments from the editor and the three reviewers. In the updated version, now we have taken into account all the suggestions which in our humble opinion substantially enhance the manuscript quality, including a careful check of the English. Taking into account the valuable suggestions provided we have included now our specific answers in the following table. Additionally, we have highlighted with yellow text in the updated manuscript the additions or changes we have made to answer the editor´s and reviewers’ requests. Thanks a lot for your professionalism dealing with our manuscript.

The lines mentioned correspond to the PDF version received from reviewers 1 and 2 in the attached file. 

Reviewer 3 Report

Comments and Suggestions for Authors

As stated in the manuscript, knowledge of the diversity and distribution of Mexican fungi is far from complete.  Focusing on boletes makes strong sense as they are ecologically, economically, and socially important.  Additionally, past studies in temperate regions of Mexico provided a framework for the work. 

This is a straightforward taxonomic paper that describes one new genus and six new species based on morphological and sequence data.  Phylogenetic analyses were employed to investigate relationships among the taxa.  There are a few typographic errors and some awkward sentences or word choices scattered throughout the manuscript, but they are not disruptive and the  manuscript is easy to read and comprehend. 

I only had two questions: (1) the Cacaoporus phylogeny is based on TEF1-ATP6 sequences, but in the Materials and Methods section they are not listed as markers that were amplified and no information on the primers used were provided. (2) All of the descriptions except Leccinum oaxacanum provide a paragraph on Habitat and Distribution.  Since related species were found associated with Ericaceae, it would be good to know the habitat and potential tree associations of this new species

Comments on the Quality of English Language

There are a few typographic errors and some awkward sentences or word choices scattered throughout the manuscript, but they are not disruptive and the manuscript is easy to read and comprehend. 

Author Response

(The authors gave the same response as above.)
